# Interaction-selective molecular sieving adsorbent for direct separation of ethylene from senary $C_2$-$C_4$ olefin/paraffin mixture

Yong Peng[1], Hanting Xiong[1], Peixin Zhang [1], Zhiwei Zhao[1], Xing Liu[1], Shihui Tang[1], Yuan Liu[1], Zhenliang Zhu[1], Weizhen Zhou[1], Zhenning Deng[1], Junhui Liu[1], Yao Zhong [1], Zeliang Wu[1], Jingwen Chen [1], Zhenyu Zhou[1], Shixia Chen[1], Shuguang Deng [2] & Jun Wang [1] ✉

Olefin/paraffin separations are among the most energy-intensive processes in the petrochemical industry, with ethylene being the most widely consumed chemical feedstock. Adsorptive separation utilizing molecular sieving adsorbents can optimize energy efficiency, whereas the size-exclusive mechanism alone cannot achieve multiple olefin/paraffin sieving in a single adsorbent. Herein, an unprecedented sieving adsorbent, BFFOUR-Cu-dpds (BFFOUR = $BF_4^-$, dpds = 4,4'-bipyridinedisulfide), is reported for simultaneous sieving of $C_2$-$C_4$ olefins from their corresponding paraffins. The interlayer spaces can be selectively opened through stronger guest-host interactions induced by unsaturated C = C bonds in olefins, as opposed to saturated paraffins. In equimolar six-component breakthrough experiments ($C_2H_4$/$C_2H_6$/$C_3H_6$/$C_3H_8$/n-$C_4H_8$/n-$C_4H_{10}$), BFFOUR-Cu-dpds can simultaneously divide olefins from paraffins in the first column, while high-purity ethylene ( > 99.99%) can be directly obtained through the subsequent column using granular porous carbons. Moreover, gas-loaded single-crystal analysis, in-situ infrared spectroscopy measurements, and computational simulations demonstrate the accommodation patterns, interaction bonds, and energy pathways for olefin/paraffin separations.

Separations are indispensable in the chemical industry, where thermal-driven processes are predominant ( > 80%). These established processes, *e.g.*, cryogenic distillations, consume a significant amount of energy due to the similar properties of gas components[1–3]. Separations of olefins and paraffins are highly important, over 350 million tons of olefins, *i.e.*, ethylene ($C_2H_4$) and propylene ($C_3H_6$), are produced mainly by steam cracking of naphtha[1,4,5]. But the separations of olefins/paraffins are incredibly challenging owing to their close relative volatilities and boiling points raised by the subtle molecular differences in single/double carbon bonds[6–8]. The dehydrogenation reactions of paraffins (*i.e.*, propane and ethane) typically exhibit a conversion yield from 50% to 60%, giving a complex mixture of $C_2$-$C_4$ olefins and paraffins[9,10]. The prevalent high-pressure cryogenic distillations for producing polymer-grade ( > 99.95%) olefins consume ~7.3 GJ per ton $C_2H_4$ and ~12.9 GJ per ton $C_3H_6$, which collectively consume 0.3% of the global energy production[11,12]. In contrast, physical adsorptions utilizing solid adsorbents with different affinities toward guest molecules enable energy-efficient olefin/paraffin separations and purifications[8,13–16].

The molecular-sieving mechanism can achieve complete separation with theoretically infinite selectivity[17]. The predominant molecular-sieving adsorbents function by differentiating molecule

[1]School of Chemistry and Chemical Engineering, Nanchang University, Nanchang, Jiangxi 330031, China. [2]School for Engineering of Matter, Transport and Energy, Arizona State University, Tempe, AZ 85287, USA. ✉e-mail: jwang7@ncu.edu.cn

sizes, that is, adsorbing smaller components while excluding larger ones (Fig. 1a)[18–21]. Therefore, the adsorbents executing the size-selective mechanism require precise control over pore sizes and apertures[22–25]. For instance, given the subtle size difference in kinetic diameter of $C_2H_4$ and $C_2H_6$ (0.028 nm in kinetic diameter, Supplementary Fig. 1), only three exceptional adsorbents (i.e., UTSA-280, M-gallate (M = Mg, Co), and HOF-FJU-1) have successfully achieved the complete exclusion of $C_2H_6$ from $C_2H_4/C_2H_6$ mixtures[10,17,26]. Although $C_3H_6$ and $C_3H_8$ have a larger size difference (0.042 nm in kinetic diameter), their structural dissimilarity is even more minor due to the same alkyl end (-$CH_3$ group), making the $C_3H_6$ sieving from $C_3H_8$ more challenging[11,27,28]. Therefore, the specific design and tuning of pore sizes are prerequisite for each olefin/paraffin pair. Note that achieving simultaneous molecular sieving of multiple olefin/paraffin pairs, such as $C_2H_4/C_2H_6$ and $C_3H_6/C_3H_8$, is theoretically impossible in rigid adsorbents.

In addition to the combination of metal ions/clusters and organic ligands in metal-organic frameworks (MOFs), the incorporation of anionic pillars as third components (e.g., $SiF_6^{2-}$, $GeF_6^{2-}$, $SO_4^{2-}$) into hybrid MOF adsorbents allows for facile tuning of pore apertures/functionalities, holding particular promise for the separations of light hydrocarbons[29–32]. Generally, the linear organic linkers (e.g., 4,4'-bipyridine, pyrazine, and 1,4-bis(4-pyridyl)benzene) and transit metal ions (e.g., $Cu^{2+}$, $Co^{2+}$, and $Zn^{2+}$) form rigid two-dimensional (2D) layers with coordination bonds, which are further coordinately connected by anion pillars to create three-dimensional (3D) frameworks (Supplementary Fig. 2a)[33–35]. Structural rigidity is typically maintained in this type of adsorbent, while the rotation of anionic pillars may offer a degree of flexibility[36–39]. Recently, our group reported a series of rigid-flexible MFSIX-Cu-dps (dps = 4,4'-dipyridyridylsulfide, M = Si, Ge, Ti), the bent dps linker created rigid intralayer cages and flexible interlayer spaces (Supplementary Fig. 2b). The selective expansion of interlayer spaces facilitated efficient $C_2H_2/C_2H_4$ and $C_2H_2/CO_2$ sieving separations[36,40]. Therefore, we expect that the judicious selection of anion pillars can decouple 2D interlayers and provide additional

dimensions of flexibility (Supplementary Fig. 2c). This intriguing feature may offer an opportunity for intricate $C_2$-$C_4$ olefin/paraffin sieving separations by distinguishing the saturation degree of carbon bonds in a single adsorbent (Fig. 1a).

Herein, we report a molecular sieving adsorbent, BFFOUR-Cu-dpds (BFFOUR = $BF_4^-$, dpds = 4,4'-bipyridinedisulfide), for simultaneous sieving of $C_2$-$C_4$ olefins ($C_2H_4$, $C_3H_6$, and n-$C_4H_8$) from their corresponding paraffins ($C_2H_6$, $C_3H_8$, and n-$C_4H_{10}$). The assembly of adjacent one-dimensional (1D) Cu(dpds)$_2$(BF$_4$)$_2$ chains are interwoven by angular $BF_4^-$ anions forming flexible interlayer spaces. The adsorption data demonstrates that $C_2$-$C_4$ olefins can expand interlayer spaces, while paraffins are excluded. Additionally, benzene exhibits selective opening of contracted spaces instead of cyclohexane, further confirming its selectivity originating from distinguishing guest-host interactions. Note that high-purity $C_2H_4$ (>99.99%) can be directly obtained through a two-column configuration from six-component $C_2$-$C_4$ olefins and paraffins mixtures. The first BFFOUR-Cu-dpds column efficiently separates olefins and paraffins, while high-purity $C_2H_4$ can be directly obtained from the second column filled with granular porous carbons. Guest-loaded single-crystal analysis reveals the structure evolution and identifies the adsorption sites. Computational simulations demonstrate that olefins exhibit significantly lower energy barriers for interlayer expansion than paraffins.

## Results
### Structure and porosity analysis
The reaction of dpds ligand and Cu(BF$_4$)$_2$ in methanol/water solution yielded purple cuboid BFFOUR-Cu-dpds (Cu(dpds)$_2$(BF$_4$)$_2$·2H$_2$O) crystals under mild conditions (Supplementary Figs 3, 4). Single-crystal X-ray diffraction (SCXRD) revealed that the as-synthesized BFFOUR-Cu-dpds crystallizes in the monoclinic Ccc2 space group (Supplementary Table 1). The distorted octahedral Cu atom was coordinated by four N atoms from four independent dpds linkers and two F atoms from two trans BF$_4^-$ anions (Supplementary Fig. 5), which was connected by dpds ligands forming infinite 1D [Cu(dpds)$_2$(BF$_4$)$_2$]$_n$ chains

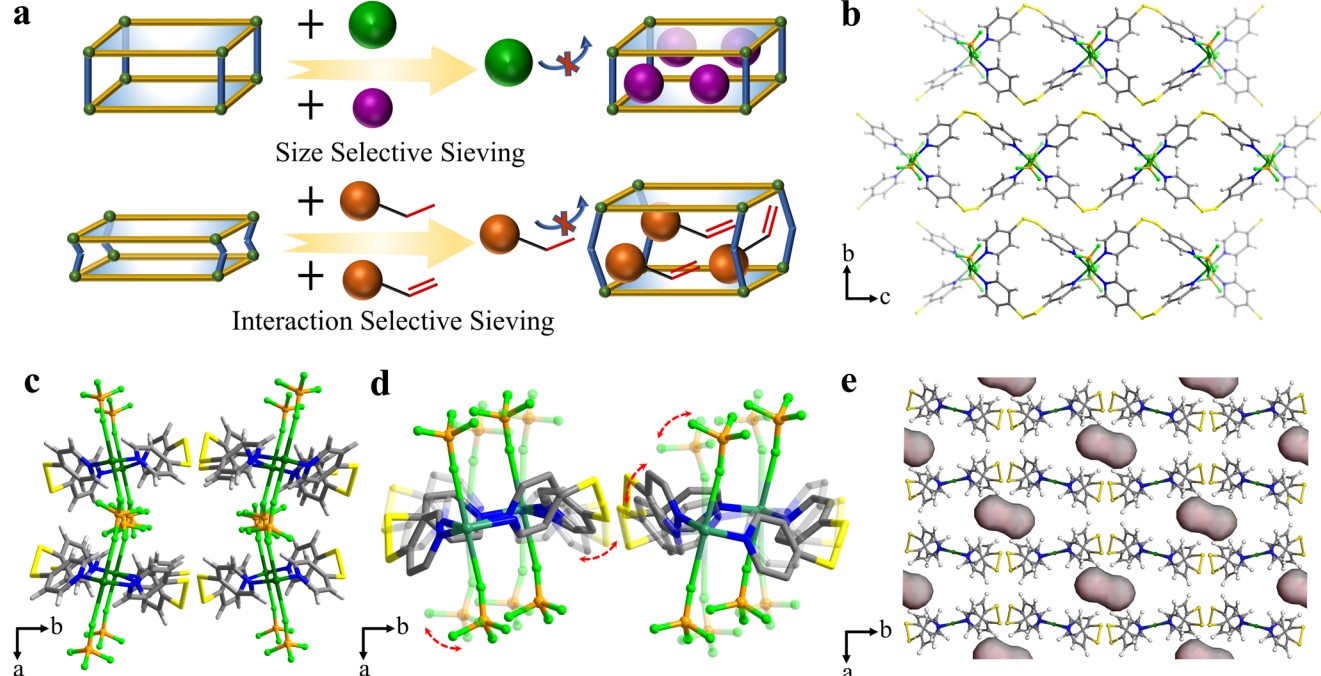

**Fig. 1 | The structure and pore properties of BFFOUR-Cu-dpds. a** Schematic of size-selective and interaction-selective molecular sieving mechanisms. **b** 1D [Cu(dpds)$_2$(BF$_4$)$_2$]$_n$ chains viewed along c axis. **c** 2D flexible layers interwoven by angular BF$_4^-$ anions via hydrogen bonds. **d** The responsive moving of 2D BFFOUR-

Cu-dpds layers. **e** The pores of activated BFFOUR-Cu-dpds viewed along c axis. Color code: Cu dark green, B orange, F light green, C dark grey, H light grey, S yellow, N blue.

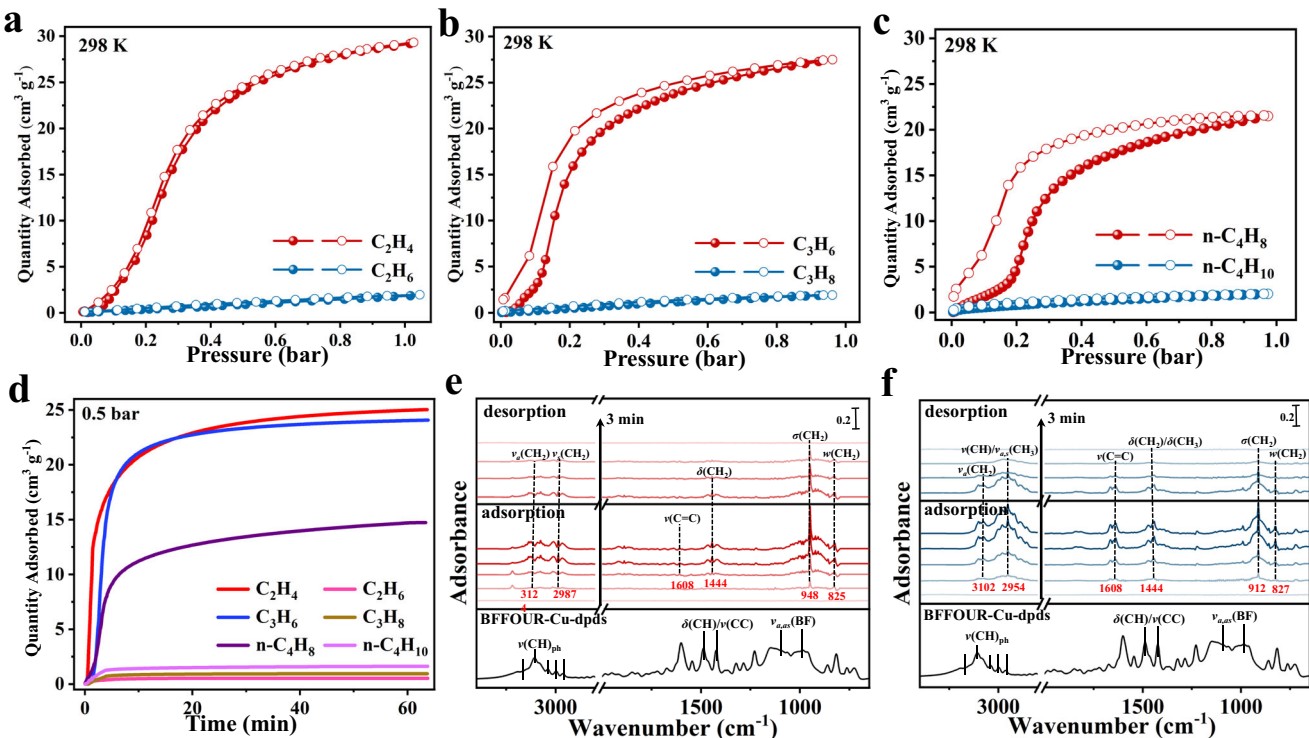

**Fig. 2 | Adsorption and separation properties of BFFOUR-Cu-dpds.** Adsorption isotherms for (**a**) $C_2H_4$ and $C_2H_6$, (**b**) $C_3H_6$ and $C_3H_8$, and (**c**) n-$C_4H_8$ and n-$C_4H_{10}$ at 298 K. **d** Kinetic adsorption curves for $C_2$-$C_4$ olefins and paraffins at 0.5 bar and 298 K. Time-dependent in-situ FT-IR spectra for **e** $C_2H_4$ and **f** $C_3H_6$ in BFFOUR-Cu-dpds.

(Fig. 1b). Adjacent 1D chains were coupled and stabilized through hydrogen bonds mediated by guest water molecules with an O···H distance of 2.68 Å (Supplementary Fig. 6). The angular $BF_4^-$ anions further interweaved the assembled flexible 2D layers via hydrogen bonds along *a* axis (Fig. 1c). Note that the generated 2D layers exhibited non-rigid behaviors and functioned as hinges with a certain degree of flexibility (Fig. 1d). Unlike rigid coordination bonds in $MF_6$-pillared MOFs, the flexible hydrogen bonds created unparallel stackings with a dihedral angle of 64.39° along *a* axis, thereby imparting additional flexibility (Supplementary Fig. 7). The asymmetric assembly resulted in discrete pore systems with an accessible pore size of 3.9 × 4.0 Å (Supplementary Fig. 8). The single-crystal to single-crystal transition was achieved after the removal of guest water molecules (Supplementary Fig. 4b). The SCXRD analysis disclosed that the activated BFFOUR-Cu-dpds $[Cu(dpds)_2(BF_4)_2]$ retained the same space group but with a slightly reduced pore volume of 2816.3(16) Å³ (Supplementary Fig. 9 and Supplementary Table 1). The enlarged dihedral angle of 76.83° reduced the accessible pore size to 3.3 × 3.8 Å (Fig. 1e and Supplementary Fig. 7b), smaller than the cross-sectional sizes of $C_2$-$C_4$ olefins and paraffins. In the activated BFFOUR-Cu-dpds, the interlayer distance was measured to be 3.55 Å (Supplementary Fig. 6b), longer than that of SiFSIX-Cu-dps (3.35 Å) and ZUL-100 (3.35 Å), indicating increased flexibility[40,41].

The phase purity of bulk BFFOUR-Cu-dpds was confirmed by powder X-ray diffraction (PXRD) patterns that matched well with the simulated one (Supplementary Fig. 10). After activation, there was no discernible alteration in the diffraction peaks. Thermogravimetric analysis revealed that BFFOUR-Cu-dpds was thermally stable up to 240 °C (Supplementary Fig. 11). Fourier transform infrared (FT-IR) spectrometer was conducted to confirm the constituents of BFFOUR-Cu-dpds (Supplementary Fig. 12). The B-F stretching vibrations at 1128.23 cm⁻¹ ($v_s$) and 1060.24 cm⁻¹ ($v_{as}$) were attributed to the $BF_4^-$ pillars[42], while the small sharp peak at 659.53 cm⁻¹ was assigned to the Cu-F bond stretching vibration[43]. The bands of C-S and S-S bonds were

observed at 533.24 cm⁻¹ and 522.61 cm⁻¹, respectively[44]. The Cu-N bonds that appeared at 499.47 and 439.69 cm⁻¹ confirmed the coordination pattern of $Cu^{2+}$ ions[45]. Furthermore, the PXRD characteristic peaks and BET specific surface areas measured at 195 K of BFFOUR-Cu-dpds almost remained intact after soaking in various organic solvents and acidic/basic aqueous solutions (pH = 5–11) for 7 days, indicating its excellent structure and chemical stability (Supplementary Fig. 13). The single crystals could also be retained after immersing in various organic solvents (Supplementary Fig. 14). The porous properties were evaluated by $CO_2$ adsorption isotherm at 195 K, the activated BFFOUR-Cu-dpds exhibited a Brunauer-Emmentt-Teller (BET) specific surface area of 140 m² g⁻¹ with a pore size centered at 3.8 Å (Supplementary Fig. 15). The experimental total pore volume was calculated to be 0.06 cm³ g⁻¹, which is slightly lower than the value obtained through SCXRD analysis (0.078 cm³ g⁻¹). Meanwhile, the contracted pores resulted in negligible $N_2$ adsorption at 77 K.

### Equilibrium and kinetic gas adsorption behaviors

Single-component equilibrium adsorption isotherms of $C_2$-$C_4$ olefins and paraffins were collected at different temperatures (273–323 K). Fig 2a-c illustrated the step-wise adsorption behavior of $C_2$-$C_4$ olefins, which can be attributed to the structural adaptability of flexible MOFs under guest stimuli[30,37,39,46,47]. At 1.0 bar, the $C_2H_4$ uptake reached 31.43 cm³ g⁻¹ and 29.31 cm³ g⁻¹ at 273 K and 298 K, respectively (Fig. 2a and Supplementary Fig. 16a). Similarly, the adsorption capacity for $C_3H_6$ was measured to be 33.90 and 27.50 cm³ g⁻¹ at 273 K and 298 K, respectively (Fig. 2b and Supplementary Fig. 17a). Meanwhile, BFFOUR-Cu-dpds could adsorb 21.49 cm³ g⁻¹ n-butene (n-$C_4H_8$) at 298 K, which increased to 24.54 cm³ g⁻¹ at 283 K (Fig. 2c and Supplementary Fig. 18a). Note that $C_2$-$C_4$ paraffins, *i.e.*, $C_2H_6$, $C_3H_8$, and n-butane (n-$C_4H_{10}$), were completely excluded by activated BFFOUR-Cu-dpds even at higher temperatures (Supplementary Fig. 19). Correspondingly, based on the experimental pore volume, the packing density of $C_2H_4$, $C_3H_6$, and n-$C_4H_8$ in BFFOUR-Cu-dpds achieved

471.47 g L$^{-1}$, 653.92 g L$^{-1}$, and 680.51 g L$^{-1}$ at 298 K and 1.0 bar, which was 414.3, 383.1, and 298.8 times higher than the density of gaseous $C_2H_4$ (1.138 g L$^{-1}$), $C_3H_6$ (1.707 g L$^{-1}$), and n-$C_4H_8$ (2.276 g L$^{-1}$) under similar conditions[17,48]. The time-dependent kinetic adsorption curves at 0.5 bar revealed an abrupt uptake point in <1 min for $C_2$-$C_4$ olefins and quickly reached equilibrium at ~13 min (Fig. 2d). The kinetic adsorption capacity of $C_2H_4$ (25.0 cm$^3$ g$^{-1}$), $C_3H_6$ (24.1 cm$^3$ g$^{-1}$), and n-$C_4H_8$ (14.7 cm$^3$ g$^{-1}$) was in good agreement with their corresponding equilibrium adsorption capacities. Meanwhile, no noticeable adsorption uptakes were observed on $C_2$-$C_4$ paraffins even after a prolonged period of ~70 mins. The kinetic adsorption curves at 1.0 bar showed a similar phenomenon (Supplementary Fig. 20).

The threshold pressure was measured to be 0.103 bar for $C_2H_4$, 0.107 bar for $C_3H_6$, and 0.157 bar for n-$C_4H_8$ at 298 K (Supplementary Fig. 21). As the temperature decreased to 273 K, the threshold pressure correspondingly reduced to 0.03 bar for $C_2H_4$, 0.028 bar for $C_3H_6$, and 0.039 bar for n-$C_4H_8$ (283 K). Intriguingly, the occurrence of similarities in threshold pressures and adsorption capacities of $C_2$-$C_4$ olefins, particularly for $C_2H_4$ and $C_3H_6$, was rare among reported flexible adsorbents[49,50]. The adsorption behaviors suggested that the adsorption processes were initiated by opening the pore channels and subsequently fulfilling the voids, rather than relying on molecule-size sieving and adsorption affinities. Therefore, we anticipated that the simultaneous molecular sieving for $C_2$-$C_4$ olefin/paraffin originated from the selective interlayer expansion driven by differences in guest-host interactions. As a proof-of-concept, we also measured the adsorption isotherms for benzene and cyclohexane (Supplementary Fig. 22a). As anticipated, benzene molecules possessing delocalized π-bonds could open the interlayer spaces, whereas the saturated structure of cyclohexane could not enter BFFOUR-Cu-dpds. Additionally, both styrene and ethylbenzene could be adsorbed by BFFOUR-Cu-dpds, indicating that carbon double bonds and benzene rings were capable of opening up interlayer spaces (Supplementary Fig. 22b).

## Separation selectivity and adsorption enthalpy

Ideal adsorbed solution theory (IAST) was applied to estimate the separation selectivity for $C_2$-$C_4$ olefins/paraffins (0.5/0.5, v/v). The dual-site Langmuir-Freundlich (DLSF) model was employed to fit the adsorption isotherms with remarkable precision (Supplementary Fig. 23 and Supplementary Table 2). Due to the stepwise adsorption behaviors, the IAST selectivity curves of $C_2$-$C_4$ olefins/paraffins exhibited an increasing trend along the increase of adsorption amounts (Supplementary Fig. 24)[51]. Specifically, BFFOUR-Cu-dpds showed a high IAST selectivity for $C_2H_4$/$C_2H_6$ (68.8), $C_3H_6$/$C_3H_8$ (108.4), and n-$C_4H_8$/n-$C_4H_{10}$ (22.9) at 298 K and 1.0 bar, surpassing many leading adsorbents such as Ni-gallate (16.8 for $C_2H_4$/$C_2H_6$), NOTT-300 (48.7 for $C_2H_4$/$C_2H_6$), ZnAtzPO$_4$ (12.4 for $C_2H_4$/$C_2H_6$), Fe$_2$(m-dobdc) (60 for $C_3H_6$/$C_3H_8$), and Fe$_2$(dobdc) (14.7 for $C_3H_6$/$C_3H_8$)[26,52–55]. To avoid the overestimation of separation performances in molecular-sieving adsorbents by IAST calculations, an intuitive evaluation based on the olefin-to-paraffin uptake ratio was employed[23,56–59]. As depicted in Supplementary Fig. 25, the uptake ratio reached 14.88 for $C_2H_4$/$C_2H_6$, 14.35 for $C_3H_6$/$C_3H_8$, and 10.53 for n-$C_4H_8$/n-$C_4H_{10}$, also outperforming most top-ranking adsorbents, including HIAM-301 (11.47 for $C_3H_6$/$C_3H_8$), Co-gallate (10.87 for $C_2H_4$/$C_2H_6$) and JNU-3 (1.21 for $C_3H_6$/$C_3H_8$), NOTT-300 (5.03 for $C_2H_4$/$C_2H_6$)[11,27,48,52]. Although the scarcity of reported data, the uptake ratio for n-$C_4H_8$/n-$C_4H_{10}$ also surpassed monolayer AgNO$_3$/SiO$_2$ sorbent (8.33) and Ag$^+$ ion impregnated clay (2.97)[60,61]. To the best of our knowledge, BFFOUR-Cu-dpds represents the example of simultaneous sieving of $C_2$-$C_4$ olefins and paraffins.

The potential co-adsorption of paraffin molecules in gas-mixtures may arguably compromise the sieving efficiency of olefins/paraffins upon expanding interlayer spaces. Thus, the binary-component adsorption isotherms for $C_2$-$C_4$ olefin/paraffin (0.5/0.5, v/v) were collected (Supplementary Fig. 26). The proximity of uptakes and threshold pressures between olefin/paraffin mixtures and their single-component olefin isotherms indicated the preferential adsorption of olefins over paraffins, even in equimolar gas mixtures. Moreover, the kinetic adsorption curves and capacities for equimolar olefin/paraffin gas-mixtures at 1.0 bar also matched well with these of pure olefin components (Supplementary Fig. 27), e.g., 24.96 cm$^3$ g$^{-1}$ for $C_2H_4$/$C_2H_6$ (0.5/0.5, v/v, 1 bar) and 24.21 cm$^3$ g$^{-1}$ for $C_2H_4$ (0.5 bar), further confirming the negligible co-adsorption and solid molecular sieving effect.

After fitting $C_2$-$C_4$ olefins adsorption isotherms by the virial equation (Supplementary Fig. 28 and Supplementary Table 3), the isosteric heat of adsorption ($Q_{st}$) was calculated to be 34.44, 31.13, and 33.37 kJ mol$^{-1}$ at near-zero coverage for $C_2H_4$, $C_3H_6$, and n-$C_4H_8$, respectively (Supplementary Fig. 29). These adsorption enthalpy values were notably lower than many leading olefin-selective adsorbents (Supplementary Table 4 and Supplementary Fig. 5), such as PAF-1-SO$_3$Ag (106 kJ mol$^{-1}$ for $C_2H_4$), Cu$^I$@UiO-66-(COOH)$_2$ (48.5 kJ mol$^{-1}$ for $C_2H_4$), Co-gallate (41 kJ mol$^{-1}$ for $C_3H_6$), and KAUST-7 (57.4 kJ mol$^{-1}$ for $C_3H_6$), suggesting the easy regenerations for BFFOUR-Cu-dpds[8,11,62,63]. Moreover, the time-dependent in-situ FT-IR spectra was carried out to demonstrate the adsorption and desorption processes. As shown in Fig. 2e, $C_2H_4$ molecules were rapidly adsorbed within 3 min, as evidenced by the appearance of stretching vibrations of $v_{a,as}(CH_2)$ at 2987 ~ 3124 cm$^{-1}$, $v(C=C)$ at 1608 cm$^{-1}$, $\delta(CH_2)$ at 1444 cm$^{-1}$, $\sigma(CH_2)$ at 948 cm$^{-1}$, and $w(CH_2)$ at 825 cm$^{-1}$ of adsorbed $C_2H_4$[14,64–66]. The intensity of these vibrations gradually escalated until reaching equilibrium at 9 min. It should be noted that BFFOUR-Cu-dpds can be facilely regenerated under mild conditions of 10 mL min$^{-1}$ He flows at 60 °C within 12 min, as evidenced by the disappeared vibrations of adsorbed $C_2H_4$. Similarly, the adsorption equilibrium of $C_3H_6$ was attained within 9 min in BFFOUR-Cu-dpds and could be easily regenerated under the same conditions within 12 min (Fig. 2f)[64,67].

## Gas-loaded single-crystal structures

Single-crystal to single-crystal transformations were attained upon $C_2H_4$ and $C_3H_6$ loadings (Supplementary Fig. 30). Despite multiple attempts, it was not able to obtain n-$C_4H_8$-loaded single-crystals. The SCXRD analysis of gas-loaded BFFOUR-Cu-dpds at 173 K revealed that olefin molecules were accommodated in the interlayer spaces as the preferential binding sites (Fig. 3a, b). Upon $C_2H_4$ adsorption, the B-F-Cu dihedral angle decreased from 166.82° to 152.53° to adapt $C_2H_4$ molecules, leading to the reduction of unit cell volume from 2816.3 Å$^3$ to 2780.4 Å$^3$ (Fig. 3E and Supplementary Fig. 31a). Due to the larger size of $C_3H_6$, the corresponding B-F-Cu dihedral angle was further reduced to 143.9° with a unit cell volume of 2801.4 Å$^3$ (Fig. 3e and Supplementary Fig. 31b). Meanwhile, the pyridine rings underwent rotation to accommodate gas molecules, the dihedral angle between the pyridine plane and metal plane decreased from 68.35° to 67.63° and 66.54° on $C_2H_4$- and $C_3H_6$-loaded BFFOUR-Cu-dpds, respectively (Supplementary Fig. 32). Similarly, another dihedral angle decreased from 44.86° to 43.11° and 40.52° on BFFOUR-Cu-dpds upon $C_2H_4$- and $C_3H_6$-loading, demonstrating its structural self-adaptation in response to gas adsorptions.

The binding conformations of $C_2H_4$ and $C_3H_6$ within BFFOUR-Cu-dpds framework were probed through SCXRD analysis, revealing multiple interactions between the olefins and interlayer spaces. For instance, each $C_2H_4$ molecule was associated with two $BF_4^-$ anions via forming strong C-H•••F interactions, in which the negatively charged F atom interacted with weakly acidic H atoms of $C_2H_4$ at bonding distances of 2.65 Å, 2.75 Å, and 2.84 Å (Fig. 3c). Moreover, two intermolecular interactions via C-H•••π ($C_2H_4$) interactions with distances of 3.03 Å and 3.09 Å implied the tight contacts between $C_2H_4$ and BFFOUR-Cu-dpds. Similarly, each $C_3H_6$ molecule formed strong C-H•••F bindings (1.97 Å, 2.73 Å and 3.01 Å) with F atoms of two $BF_4^-$ ions, as well as two C-H-π ($C_3H_6$) interactions (2.52 Å and 2.63 Å) with BFFOUR-Cu-dpds (Fig. 3d). The binding sites of n-$C_4H_8$ were investigated by computational simulations, and a detailed discussion will follow.

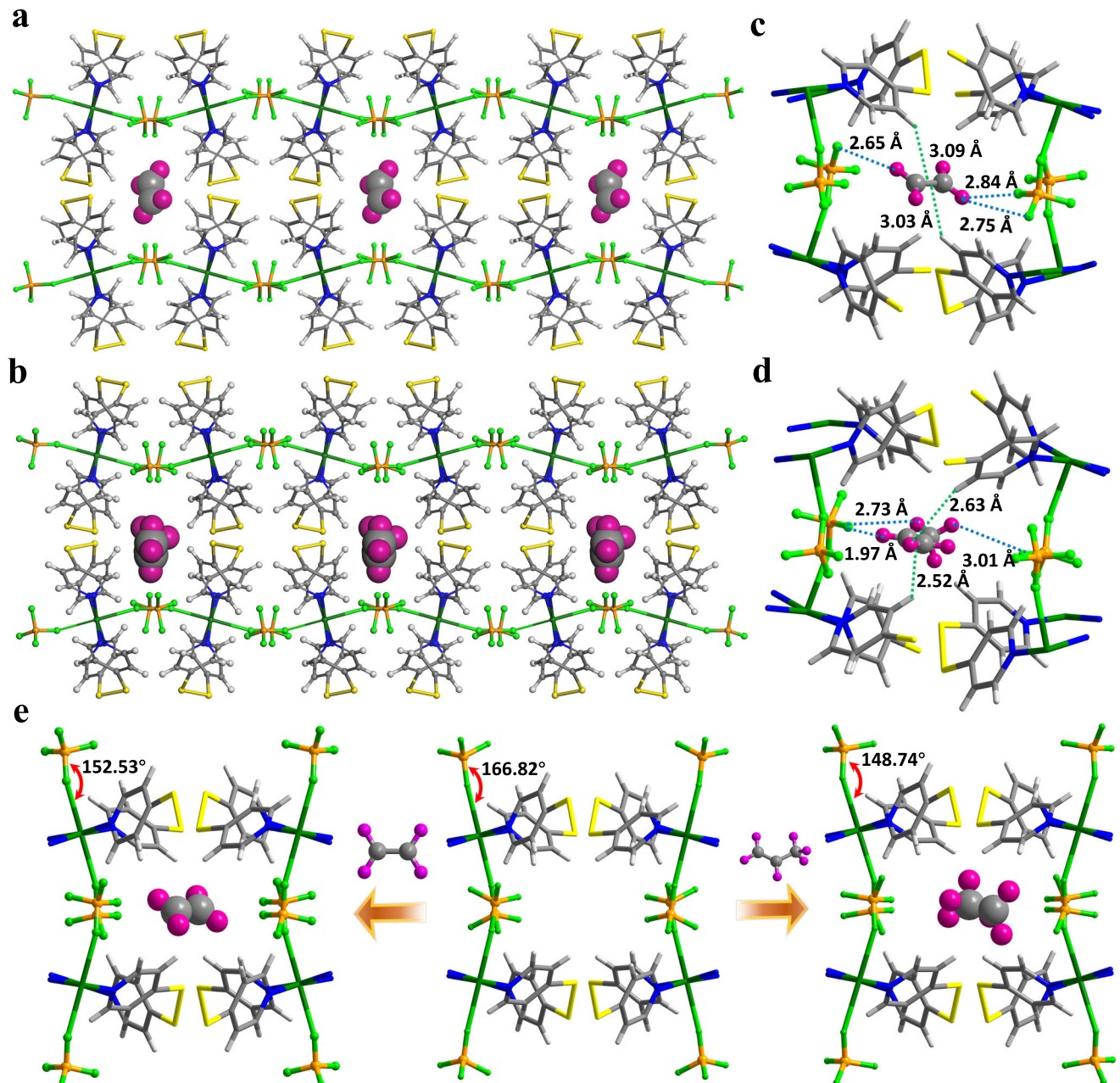

**Fig. 3 | The crystal structure of gas-loaded BFFOUR-Cu-dpds. (a)** $C_2H_4$-loaded and **(b)** $C_3H_6$-loaded BFFOUR-Cu-dpds viewed along *c* axis. Adsorption sites for **(c)** $C_2H_4$ and **(d)** $C_3H_6$ derived from SCXRD analysis of gas-loaded BFFOUR-Cu-dpds. **e** Structure change of BFFOUR-Cu-dpds framework after gas loadings. Color code: F green, Cu dark green, C gray, H (in framework) light gray, H (in hydrocarbons) dark purple, N blue, S yellow, blue line: C-H•••F interaction, green line: C-H•••π (olefin) binding.

## Modeling simulation studies

To elucidate interaction-selective adsorption behaviors, molecular simulations were conducted to illustrate the energy changes during corresponding transition states. The diffusion process of gas molecules into the BFFOUR-Cu-dpds framework can be rationally divided into four distinct transit states[36]. In the initial state (State I), the optimized framework and gas molecules were established as independent energy references. Subsequently, upon adsorption onto the surface of BFFOUR-Cu-dpds (State II), the gas molecules further diffused into pore channels (State III) and were firmly captured by adsorption sites (State IV). In terms of energy, the structural transformation during the diffusion process from State II to State III required additional energy input, representing the highest energy level in host-guest systems. Whereas, State IV was considered as having the minimal

conformational energy landscape. Therefore, the energy difference for mitigating gas molecules from the adsorbent surface into the pore channel (State II to State III) was denoted as the energy barrier (ΔE′) for structural deformation and guest accumulation. In other words, it was a quantitative measurement of the permeation ease of the probe molecule into pore channels[68,69].

In particular, $C_2H_4$, $C_3H_6$, and n-$C_4H_8$ exhibited comparable ΔE′ values of 0.502 eV, 0.448 eV, and 0.494 eV, respectively (Fig. 4a-c), which was consistent with their close threshold pressures as shown in the adsorption isotherms. In stark contrast, $C_2H_6$, $C_3H_8$, and n-$C_4H_{10}$ molecules with saturated carbon bonds and large sizes exhibited significantly higher ΔE′ of 2.704 eV, 5.617 eV, and 6.748 eV, respectively (Supplementary Figs 33–35). These findings indicated the considerable energy gap in inducing structural deformation by paraffin molecules to

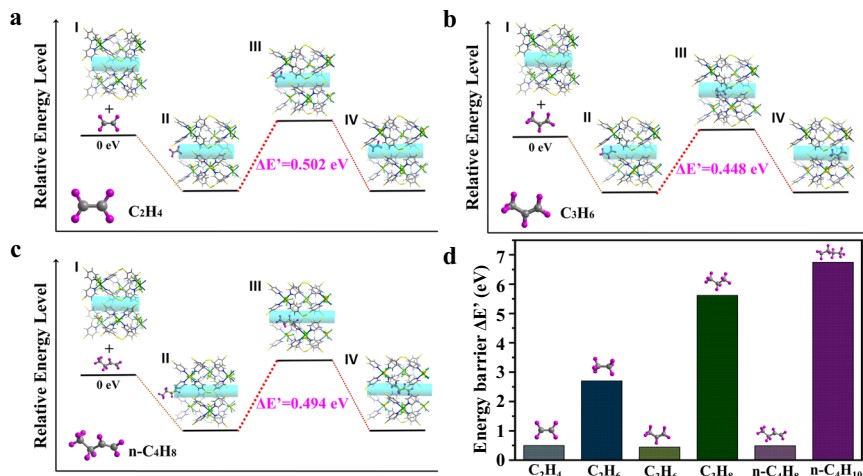

**Fig. 4 | The energy landscape of the host-guest conformations.** The energy pathway and corresponding energy levels for (**a**) $C_2H_4$, (**b**) $C_3H_6$, and (**c**) n-$C_4H_8$. **d** Comparison of energy barriers of $C_2$-$C_4$ olefins and paraffins.

compensate for the more significant transformations of paraffin-loaded frameworks (Fig. 4d).

The host-guest interactions were also illustrated through dispersion-corrected density functional theory (DFT-D) calculations. The interactions between $C_2H_4/C_3H_6$ and frameworks were almost identical with the SCXRD analysis, demonstrating the accuracy of depicting adsorption mechanisms (Supplementary Fig. 36). For the adsorption of n-$C_4H_8$, each molecule was trapped and bound to $BF_4^-$ through three cooperative C-H•••F bonds with distances ranging from 2.31 Å to 3.14 Å, as well as two cooperative C-H•••π (n-$C_4H_8$) interactions with distances of 2.5 Å and 2.62 Å (Supplementary Fig. 36c). Additionally, charge transfer analysis on the gas-loaded BFFOUR-Cu-dpds structures revealed a significant charge bias between H atoms of olefins and F atoms of $BF_4^-$ anions (Supplementary Fig. 37)[70]. The static binding energy was calculated to be 59.43 kJ mol$^{-1}$, 71.87 kJ mol$^{-1}$, and 82.52 kJ mol$^{-1}$ for $C_2H_4$, $C_3H_6$, and n-$C_4H_8$, respectively.

## Breakthrough experiments for olefin/paraffin separations

Dynamic breakthrough experiments were conducted on BFFOUR-Cu-dpds columns using binary gas-mixtures of olefin/paraffin (0.5/0.5, v/v) at 298 K to confirm its practical separation performances (Supplementary Fig. 38). As depicted in Fig. 5, BFFOUR-Cu-dpds demonstrated efficient separations of $C_2$-$C_4$ olefin/paraffin binary gas-mixtures within a single adsorption column. For the gas-mixture of $C_2H_4/C_2H_6$ (0.5/0.5, v/v), $C_2H_6$ was rapidly eluted from the column at a flow rate of 1.0 mL min$^{-1}$, while $C_2H_4$ exhibited substantial retention in the column for 28 min until saturation (Fig. 5a). Notably, the efficient $C_2H_4/C_2H_6$ separation was also obtained despite slightly decreased retention time under humid conditions (RH = 61.9%). Similarly, both $C_3H_8$ and n-$C_4H_{10}$ immediately broke through the column, whereas $C_3H_6$ and n-$C_4H_8$ were detected at retention times of 32.2 min and 21 min, respectively (Fig. 5b, c). Considering that the IAST selectivity and uptake ratio are determined by equilibrium effect, the dynamic selectivity based on breakthrough curves was calculated to be 9.16, 8.76, and 3.18 for equimolar $C_2H_4/C_2H_6$, $C_3H_6/C_3H_8$, and n-$C_4H_8$/n-$C_4H_{10}$, respectively. These values demonstrate comparable performance to top-ranking adsorbents, such as NUS-6(Hf)-Ag (4.4 for $C_2H_4/C_2H_6$)[71], ZJU-75a (14.7 for $C_3H_6/C_3H_8$)[72], Y-abtc (8.3 for $C_3H_6/C_3H_8$)[73] and KAUST-7 (12.0 for $C_3H_6/C_3H_8$)[8]. The dynamic adsorption capacity for $C_2H_4$, $C_3H_6$, and n-$C_4H_8$ were calculated to be 17.05 cm$^3$ g$^{-1}$, 19.97 cm$^3$ g$^{-1}$, and 14.43 cm$^3$ g$^{-1}$ respectively, which closely matched their static adsorption amounts at 0.5 bar. Furthermore, clean separations of $C_2H_4/C_2H_6$ and $C_3H_6/C_3H_8$ could also be achieved at higher flow rates of 2.0 and 4.0 mL min$^{-1}$ (Fig. 5a, b). Note that facile adsorbent regeneration was a

critical process to obtain high-purity olefins. After reaching the breakthrough point, the column was purged with He sweeping at 5 mL min$^{-1}$ and 333 K (Supplementary Fig. 40). The productivity of $C_2H_4$ and $C_3H_6$ with ≥99.5% purity was calculated to be 11.92 L kg$^{-1}$ and 14.19 L kg$^{-1}$ in a single adsorption-desorption cycle, which was comparable to the top-ranking adsorbents including UTSA-280 (22.08 L kg$^{-1}$ 99.2% $C_2H_4$)[17], NOTT-300 (19.66 L kg$^{-1}$ 99.2% $C_2H_4$)[52], KAUST-7 (10.7 L kg$^{-1}$ 98.3% $C_3H_6$)[8], and Co-gallate (14.9 L kg$^{-1}$ 98.7% $C_3H_6$)[11]. Meanwhile, the productivity of n-$C_4H_8$ was estimated to be 7.4 L kg$^{-1}$ with ≥90% purity (Supplementary Fig. 40f).

In addition, we introduced 20% inert He into olefin/paraffin gas-mixtures (He/olefin/paraffin, 0.2/0.4/0.4, v/v/v) to validate the absence of co-adsorption phenomenon during breakthrough experiments at a flow rate of 2.5 mL min$^{-1}$. As shown in Supplementary Fig. 41, the paraffins and He almost concurrently outflow from the column, indicating that the negligible paraffin adsorptions in gas-mixtures similar to He[74]. The dynamic adsorption capacities of $C_2H_4$, $C_3H_6$, and n-$C_4H_8$ were determined to be 15.17 cm$^3$ g$^{-1}$, 17.55 cm$^3$ g$^{-1}$, and 13.37 cm$^3$ g$^{-1}$, respectively, which were in good agreement with their corresponding static adsorption amounts (20.09 cm$^3$ g$^{-1}$ for $C_2H_4$, 22.21 cm$^3$ g$^{-1}$ for $C_3H_6$, and 15.78 cm$^3$ g$^{-1}$ for n-$C_4H_8$) at 0.4 bar.

If the separation of $C_2$-$C_4$ olefins and paraffins can be achieved in a single adsorption column, it would greatly streamline the process flow for $C_2H_4$ production (Fig. 5d). The equimolar six-component gas-mixture of $C_2$-$C_4$ olefins and paraffins ($C_2H_4/C_2H_6/C_3H_6/C_3H_8$/n-$C_4H_8$/n-$C_4H_{10}$) was introduced into the BFFOUR-Cu-dpds column at 1.0 mL min$^{-1}$ and 298 K. Surprisingly, three paraffins, i.e., $C_2H_6$, $C_3H_8$, and n-$C_4H_{10}$, were concurrently eluted from the column at the very beginning, whereas the corresponding olefins, *i.e.*, $C_2H_4$, $C_3H_6$, and n-$C_4H_8$, broke through the column at 14 min (Fig. 5e). Similarly, the clean separations for $C_2$-$C_4$ olefins can also be maintained even at a higher flow rate of 2.0 mL min$^{-1}$ (Supplementary Fig. 42). These results endow a streamlined process flow for direct $C_2H_4$ production from gas-mixtures of $C_2$-$C_4$ olefins and paraffins, if the second adsorption column could capture $C_3H_6$ and n-$C_4H_8$ during the desorption process of the first BFFOUR-Cu-dpds column (Fig. 5d and Supplementary Fig. 43)[7,28,75]. Porous carbons exhibited exceptional adsorption properties in the separation of gas molecules with varying carbon atoms with remarkable stability, cost-effectiveness, and high adsorption capacity[5,76,77]. We utilized the bamboo-derived granular carbon adsorbent (GBC-900), which was previously reported by our group, in the second adsorption column[78]. The adsorption isotherms showed steep $C_3H_8$ and n-$C_4H_{10}$ adsorptions at low pressures, whereas $C_2H_4$ was weakly adsorbed (Supplementary Fig. 44). After He sweeping for about

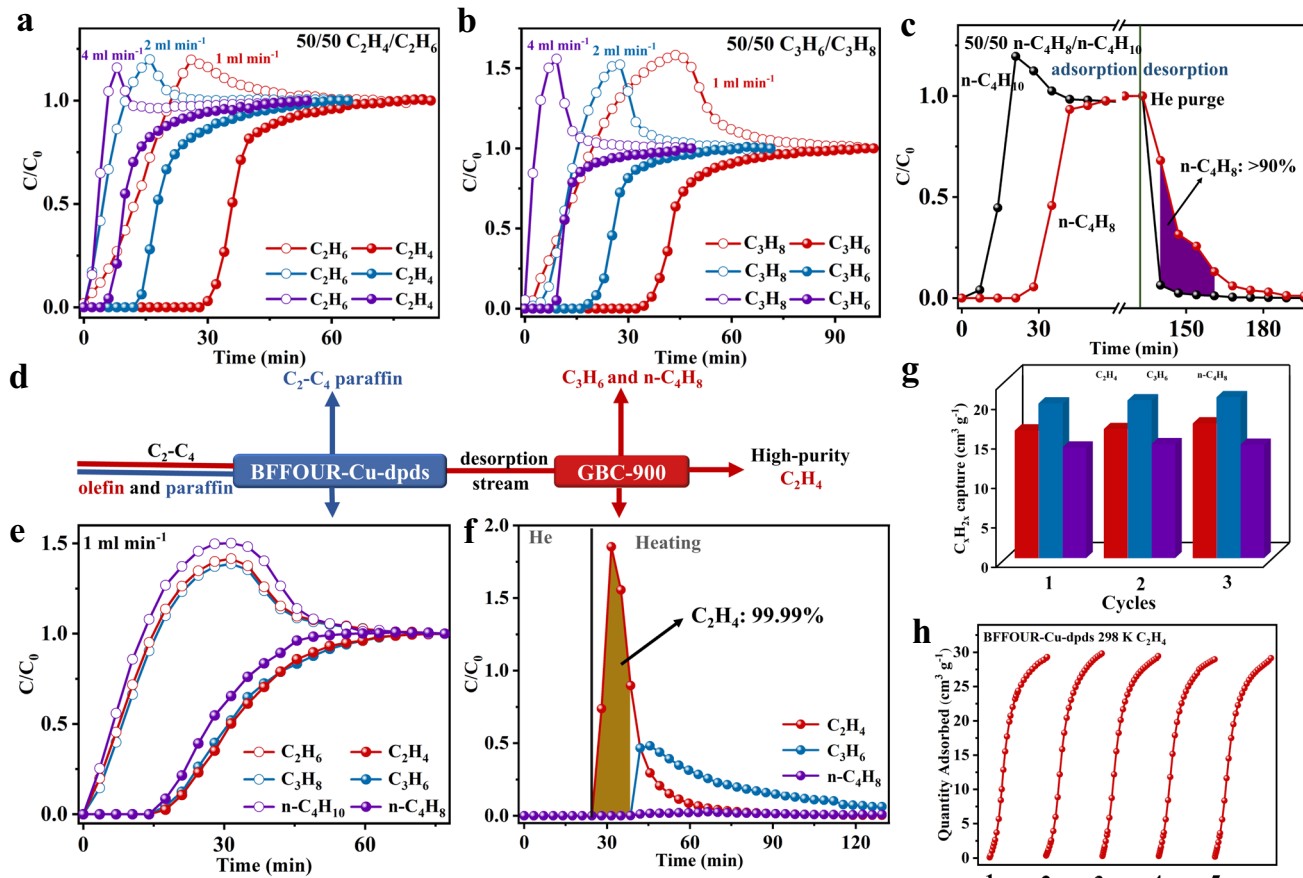

**Fig. 5 | The olefin/paraffin separation performances.** Breakthrough curves for an equimolar mixture of (**a**) $C_2H_4/C_2H_6$ (0.5/0.5, v/v) and (**b**) $C_3H_6/C_3H_8$ (0.5/0.5, v/v) at different flow rates on BFFOUR-Cu-dpds. **c** Breakthrough and desorption curves of n-$C_4H_8$/n-$C_4H_{10}$ (0.5/0.5, v/v) at 1.0 mL min$^{-1}$. **d** The schematic diagram of one-step $C_2H_4$ separation from $C_2$-$C_4$ olefin/paraffin mixture. **e** Breakthrough curve of an equimolar $C_2$-$C_4$ olefin/paraffin six-component gas-mixture at 298 K. **f** Breakthrough curve of the desorption stream from BFFOUR-Cu-dpds column in the GBC-900 column at 353 K. **g** Dynamic olefin uptakes for three consecutive breakthrough cycles. **h** Five repeated adsorption isotherms for $C_2H_4$ o at 298 K and 1.0 bar.

20 min to eliminate weakly absorbed paraffins, the BFFOUR-Cu-dpds column was heated to 353 K. The desorption stream primarily consisted of $C_2$-$C_4$ olefins with trace amounts of paraffins (Supplementary Fig. 45), which was subsequently passed through the GBC-900 column at 1.0 mL min$^{-1}$ and 353 K. Notably, $C_2H_4$ was first detected at the exit at 24.5 min, followed by $C_3H_6$ and n-$C_4H_8$ at about 38.5 min (Fig. 5f). The high-purity (>99.99%) $C_2H_4$ product could be directly collected for a duration of 14 min.

The adsorbents were anticipated to exhibit outstanding stability and reusability. Three consecutive breakthrough experiments were conducted (Supplementary Fig. 46), resulting in nearly overlapped breakthrough curves and closely matched dynamic adsorption capacities (17.05 cm$^3$ g$^{-1}$ for $C_2H_4$, 19.97 cm$^3$ g$^{-1}$ for $C_3H_6$, and 14.43 cm$^3$ g$^{-1}$ for n-$C_4H_8$, Fig. 5g). Moreover, the breakthrough experiments conducted on a six-component gas-mixture of $C_2$-$C_4$ olefins and paraffins showed almost identical curves at flow rates of 1.0 and 2.0 mL min$^{-1}$ in three successive trials (Supplementary Figs 47, 48). The recorded five repeated single-component olefin adsorption isotherms also demonstrated consistent adsorption capacities without any deterioration (Fig. 5h and Supplementary Fig. 49).

## Discussions

The BFFOUR-Cu-dpds adsorbent was successfully designed and synthesized for the simultaneous $C_2$-$C_4$ olefin sieving from their corresponding paraffin. The differences in the strength of guest-host interactions caused the selective expansion of interlayer spaces for the sieving effect rather than molecular sizes. The unique feature made BFFOUR-Cu-dpds the versatile adsorbent for multiple olefin/paraffin pairs sieving. Gas-loaded single-crystal analysis and in-situ FT-IR spectra identified the accommodation patterns and interaction bonds between olefins and adsorbent. Computational simulations revealed that the energy input of structural transformation for olefins was significantly lower than that of paraffins. Furthermore, the direct production of $C_2H_4$ from a six-component gas-mixture was achieved in a two-column separation configuration, which could significantly streamline the olefin/paraffin separation processes.

## Methods

### Materials and reagents

All chemicals were commercially available and used without any further purification, including Cupric tetrafluoroborate hydrate (Cu(BF$_4$)$_2$•xH$_2$O, Aldrich), 4,4'-dipyridyridylsulfide (dpds, 99%, Meryer), and anhydrous methanol (99%, Aladdin). All used gases, including ethylene ($C_2H_4$, 99.99%), ethane ($C_2H_6$, 99.99%), propylene ($C_3H_6$, 99.99%), propane ($C_3H_8$, 99.99%), nitrogen (N$_2$, 99.999%), helium (He, 99.999%), $C_2H_4/C_2H_6$ (0.5/0.5, v/v), $C_3H_6/C_3H_8$ (0.5/0.5, v/v), n-$C_4H_8$/n-$C_4H_{10}$ (0.5/0.5, v/v), $C_2H_4/C_2H_6/C_3H_6/C_3H_8$ (0.25/0.25/0.25/0.25, v/v/v/v) were purchased from Nanchang Jiangzhu Gas Co., Ltd (China).

### Preparation of BFFOUR-Cu-dpds

The synthesis of BFFOUR-Cu-dpds powder was as follows: 30 mL methanol solution of dpds (0.0881 g, 0.4 mmol) and an aqueous solution of Cu(BF$_4$)$_2$•xH$_2$O (0.16 g, 0.25 mmol) were mixed at room temperature, then kept undisturbed for 48 h. Subsequently, the

obtained purple powders were washed with methanol and dried under a high vacuum at 80 °C for 24 h.

The production of single crystals of BFFOUR-Cu-dpds was prepared by the static diffusion method by using a methanol solution (1.0 mL) of dpds (0.00881 g, 0.04 mmol) on an aqueous solution (1 mL) of $Cu(BF_4)_2 \cdot xH_2O$ (0.01 g, 0.07 mmol) at room temperature in a watch glass without any agitation. To control the reaction rate, 2 mL of a 1:1 methanol/$H_2O$ solution was layered between the upper and lower solutions. Light purple granular crystals were formed in one week.

## Powder and single-crystal X-ray diffraction analysis

The powder diffraction patterns were recorded in a PANalytical Empyrean Series 2 diffractometer with Cu Kα radiation, a step size of 0.0167°, a scanning time of 15 s per step, and 2θ range of 5 to 90° at ambient temperature. The single-crystal X-ray diffraction data were collected at 193(2) K using a Bruker-AXS D8 VENTURE diffractometer equipped with a PHOTON-100/CMOS detector (GaKα, λ = 1.3414 Å). The indexing was performed using APEX2, while data integration and reduction were completed using SaintPlus 6.01. Absorption correction was conducted using the multi-scan method implemented in SADABS. The space group was determined using XPREP implemented in APEX2.1. The structures were solved by direct methods and refined by nonlinear least-squares on F2 using SHELXL-97, OLEX2 v1.1.5, and WinGX v1.70.01 program packages. All non-hydrogen atoms were refined anisotropically. The contribution of disordered solvent molecules was treated using the Squeeze routine implemented in Platon.

## Thermogravimetric analysis (TGA)

The thermogravimetric analysis (TGA) data were obtained using a NETZSCH Thermogravimetric Analyzer (STA2500) under an $N_2$ atmosphere with a heating rate of 10 °C min$^{-1}$ and a temperature range of 25 to 800 °C.

## Structure stability Tests

To evaluate the solvent stability, 100 mg BFFOUR-Cu-dpds were placed separately in 20 mL vials with 15 mL of different organic and pH solvents for 7 days. The resulting solid was filtered, activated at 80 °C for 12 h, and characterized by PXRD and adsorption analysis. The used acid was sulfuric acid solution and sodium hydroxide solution with various concentrations.

## Gas adsorption measurements

A Micromeritics Three-Flex adsorption apparatus (Micromeritics Instruments, USA) was used to measure the adsorption isotherms of single-component gases ($C_2H_4$, $C_2H_6$, $C_3H_6$, $C_3H_8$) at 273 K, 298 K, and 323 K, while n-$C_4H_8$ and n-$C_4H_{10}$ were measured at 283 K, 298 K, and 313 K. To eliminate residual guest solvents in the framework, the fresh powder samples were subjected to evacuation under a high vacuum at 353 K for 24 h before each adsorption measurement. Liquid nitrogen and a dry ice-acetone bath were used for adsorption isotherms at 77 K or 195 K to calculate the Brunauer-Emmett-Teller (BET) specific surface area. The degas procedure was repeated on the same sample between measurements for 24 h.

## Kinetic adsorption measurements

Intelligent Gravimetric Analyzer (IGA-100, HIDEN) was used to measure the time-dependent adsorption profiles of olefins and paraffins. About 80 mg of BFFOUR-Cu-dpds were subjected to evacuation under a high vacuum at 353 K for 24 h before each adsorption measurement. After being cooled to a specific temperature, a single-component gas or olefin/paraffin gas mixture was introduced into the chamber, and the mass of the sample loaded with gas-molecules was continuously recorded for 80 min.

## FT- and In-situ infrared (IR) spectroscopic measurements

The IR spectra data were collected using a Bruker Tensor 27 FTIR spectrometer with a liquid $N_2$-cooled mercury cadmium telluride (MCT-A) detector. A Specac Ltd. vacuum cell (product number P/N 5850c) was placed in the sample compartment, and the BFFOUR-Cu-dpds powder (~30 mg) was placed at the focal point of the beam. As for in-situ IR spectra, the cell was connected to gas lines for $C_2H_4$ and $C_3H_6$ and a vacuum line for evacuation. The sample was first activated by heating up to 80 °C under vacuum and then cooled to 25 °C for recording the reference spectrum. The spectra data were recorded during gas adsorption saturation, and the reference spectrum was retaken after loaded gases were fully evacuated. After adsorption saturation, the olefin gas was switched to 10 mL min$^{-1}$ He for desorption at 60 °C.

## Transient breakthrough experiments

The breakthrough experiments were carried out in a homemade setup. BFFOUR-Cu-dpds (1.17 g) was packed into a Φ 6 × 200 mm stainless steel column in the sample holder. The adsorption bed was purged with a carrier gas (He ≥ 99.999%) at 353 K for 12 h and then cooled to room temperature. Then, binary $C_2H_4$/$C_2H_6$ (0.5/0.5, v/v), $C_3H_6$/$C_3H_8$ (0.5/0.5, v/v), and n-$C_4H_8$/n-$C_4H_{10}$ (0.5/0.5, v/v) gas-mixture at a flow rate of 1 mL min$^{-1}$ was separately introduced into the column at 298 K and 1 bar. Gas flows were regulated using a mass flow meter, and the outlet gas from the column was continuously monitored using gas chromatography (A91 Plus PANNA) with a flame ionization detector (FID). For the breakthrough experiments with water vapor, the gas mixture passed through a water vapor saturator under ambient conditions. The humidity of the mixture was measured by a relative humidity meter (UT333, UNI-T, China) at the inlet of the column. Following each breakthrough experiment, the sample was desorbed in-situ in the column with a He sweeping of 5 mL min$^{-1}$ at 333 K for 2 h and attained high-purity olefin gas (99.5%). To illustrate the co-adsorption problem, the required gases He/$C_2H_4$/$C_2H_6$ (0.2/0.4/0.4, v/v/v), He/$C_3H_6$/$C_3H_8$ (0.2/0.4/0.4, v/v/v), and He/n-$C_4H_8$/n-$C_4H_{10}$ (0.2/0.4/0.4, v/v/v) were mixed independently in the pipeline and introduced to the column at a total flow rate of 2.5 mL min$^{-1}$.

To further confirm its separation potential between olefin and paraffin and acquire higher purity ethylene (>99.99%), the equimolar $C_2$-$C_4$ olefin/paraffin six-component mixture was mixed in the pipeline and introduced to two columns. The first column was still primarily filled with 1.17 g BFFOUR-Cu-dpds, and then purged the trapped olefins with 1 mL min$^{-1}$ He at 353 K in BFFOUR-Cu-dpds after first column adsorption saturation. The desorption stream went through the second column filled with granular porous carbons GBC-900 at 353 K. The outlet gas from the column was monitored using gas chromatography (A91 Plus PANNA) with an FID coupled with a thermal conductivity detector (TCD). Based on the mass balance, the gas adsorption capacities can be determined as follows:

$$Q_i = \frac{C_i V}{22.4 \times m} \times \int_0^t \left(1 - \frac{F}{F_0}\right) dt \tag{1}$$

Here $Q_i$ is the dynamic adsorption capacity of gas $i$ (mmol g$^{-1}$), $C_i$ is the feed gas concentration, $V$ is the volumetric feed flow rate (cm$^3$ min$^{-1}$), t is the adsorption time (min), $F_0$ and $F$ are the inlet and outlet gas molar flow rates, and m is the mass of the adsorbent (g).

Dynamic separation selectivity (α) obtained by calculating the integral area of the desorption curves of alkanes and olefins during the blowing process. The calculation formula is as follows:

$$S_i = \int_{t1}^{t2} \left(1 - \frac{F}{F_0}\right) dt \tag{2}$$

$$\alpha = \frac{S_1/S_2}{y_1/y_2} \tag{3}$$

Here $S_i$ is the integral area of the desorption curve of gas $i$, $t_1$ and $t_2$ represent the start and end times of desorption (min), $F_O$ and $F$ are the inlet and outlet gas molar flow rates, $y_1$ and $y_2$ represent the mole fractions of 1 and 2.

## Calculation of isosteric heat of adsorption ($Q_{st}$)

The adsorption data for $C_2H_4$ and $C_3H_6$ at temperatures of 273 K, 298 K, and 323 K were used to calculate the experimental adsorption enthalpy ($Q_{st}$) and evaluate the binding strength between the adsorbent and adsorbate. For n-$C_4H_8$, the data was collected at temperatures of 283 K, 298 K, and 313 K. The viral equation was used to fit the adsorption curves:

$$\ln P = \ln N + \frac{1}{T}\sum_{i=0}^{m} a_i N^i + \sum_{i=0}^{n} b^i N^i \tag{4}$$

$$Q_{st} = -R\sum_{i=0}^{m} a_i N^i \tag{5}$$

Here $N$ is gas uptake (mg g$^{-1}$), $P$ is pressure (mmHg), a and b are virial coefficients, m and n are the coefficients required to describe the isotherm adequately.

## Ideal adsorbed solution theory (IAST) selectivity calculations

To evaluate the separation selectivity of BFFOUR-Cu-dpds for binary gas-mixtures, the dual-site Langmuir-Freundlich (DSLF) model was used to fit the single-component adsorption isotherms of $C_2H_4$, $C_2H_6$, $C_3H_6$, $C_3H_8$, n-$C_4H_8$, and n-$C_4H_{10}$. The adsorption isotherms and gas selectivity for $C_2H_4/C_2H_6$ (0.5/0.5, v/v), $C_3H_6/C_3H_8$ (0.5/0.5, v/v), and n-$C_4H_8$/n-$C_4H_{10}$ (0.5/0.5, v/v) at 298 K were calculated using the ideal adsorbed solution theory (IAST).

Dual-site Langmuir-Freundlich (DSLF) model is listed below:

$$N = N_1^{max} \times \frac{b_1 P^{c_1}}{1 + b_1 P^{c_1}} + N_2^{max} \times \frac{b_2 P^{c_2}}{1 + b_2 P^{c_2}} \tag{6}$$

Here $P$ (kPa) is the pressure of the bulk gas at equilibrium with the adsorbed phase, N (mol kg$^{-1}$) is the adsorbed amount per mass of adsorbent, $N_1^{max}$ and $N_2^{max}$ (mmol g$^{-1}$) are the saturated capacities of site 1 and site 2, $b_1$ and $b_2$ (kPa$^{-1}$) are the affinity coefficients of site 1 and site 2, and $c_1$ and $c_2$ represent the deviations from an ideal homogeneous surface.

The adsorption selectivity for the gas-mixtures of $C_2H_4/C_2H_6$ (0.5/0.5, v/v), $C_3H_6/C_3H_8$ (0.5/0.5, v/v), and n-$C_4H_8$/n-$C_4H_{10}$ (0.5/0.5, v/v) are defined by

$$S_{ads} = \frac{q_1/q_2}{p_1/p_2} \tag{7}$$

In the above equation, $q_1$ and $q_2$ are the absolute component loadings of the adsorbed phase in the mixture with partial pressures $p_1$ and $p_2$. Adsorption isotherms and gas selectivity were calculated by IAST for mixed $C_2H_4/C_2H_6$ (0.5/0.5, v/v), $C_3H_6/C_3H_8$ (0.5/0.5, v/v), and n-$C_4H_8$/n-$C_4H_{10}$ (0.5/0.5, v/v).

## Density functional theory (DFT) calculations

The first-principles DFT calculations were carried out using the CASTEP module in Materials Studio 2023[79]. The van der Waals interactions were accounted for by incorporating dispersive forces into the conventional DFT[80]. The calculations were performed under the generalized gradient approximation (GGA) with Perdew-Burke-Ernzerhof (PBE) exchange-correlation functional. A cutoff energy of 600 eV and a $2 \times 2 \times 3$ k-point mesh were used to achieve a total energy convergence of within 0.01 meV/atom. The crystal structures of the synthesized materials were optimized from the reported crystal structures. The pristine structure and an isolated gas molecule were optimized and relaxed as reference structures to calculate the binding energy. The gas molecules of $C_2H_4$, $C_3H_6$, and n-$C_4H_8$ were introduced to different locations of the channel pore, and their structures were fully relaxed. According to the output files from the CASTEP modules, the "NB dispersion corrected est. 0 K energy* (Ecor-0.5TS)" was selected as the energy at 0 K. The static binding energy (at T = 0 K) was calculated using the following equation:

$$\Delta E = E(gas) + E(adsorbent) - E(adsorbent + gas) \tag{8}$$

## The energy barrier calculation method

The energy barrier calculations were carried out using the DMol3 module in Materials Studio 2023. The periodic slab models with periodic boundary conditions were used to represent the pore surface of BFFOUR-Cu-dpds. The unit cells were optimized until the interaction force was reduced to below 0.002 Ha/Å, achieving SCF convergence of $10^{-6}$. A Global orbital cutoff of 5.1 Å was employed. The unit conversion was performed using the equation: 1 Ha = 27.212 eV. The diffusion behavior of guest molecules was investigated by calculating transition state energies through the climbing nudged elastic band (cNEB) method[81,82]. First, the surface model and host structure were optimized by employing experimentally obtained single crystal structures as initial geometries, followed by full structural relaxation. The isolated guest molecules ($C_2H_4$, $C_2H_6$, $C_3H_6$, $C_3H_8$, n-$C_4H_8$, and n-$C_4H_{10}$) were placed in a supercell and relaxed to serve as references. Subsequently, the guest molecules were introduced onto both the host surface and various locations within the pores of BFFOUR-Cu-dpds. Full structural relaxation was performed for each configuration. The optimized configurations with the lowest energy after optimization were selected for subsequent analysis and calculation. The transition state search calculations were conducted to identify transition states associated with guest transport between the two energy minimum configurations.

The optimized host-guest structures were defined as state I, and its system energy was set as the reference. The state II denoted the initial state, characterized by introducing guests onto the surface of optimized host-guest structures. State III was defined as the transition state, and state IV represented the final state where guests were pulled into the pore channel of optimized host-guest structures. Notably, $E(Initial State) = E(Final State)$. The energy barrier was determined using the following:

$$\Delta E' = E(Transition State) - E(Initial State) \tag{9}$$

Where $E(Transition State)$ is the transition energy, $E(Initial State)$ is the energy of the optimized host-guest structure where guests were introduced onto the host surface.

## Data availability

All data supporting the finding of this study are available within this article and its Supplementary Information. Crystallography information files (CIFs) of reported samples are deposited at the Cambridge Crystallographic Data Centre (CCDC, http://www.ccdc.cam.ac.uk) under reference numbers of 2162794 for as-synthesized BFFOUR-Cu-dpds, 2162081 for activated BFFOUR-Cu-dpds, 2162073 for $C_2H_4$-loaded BFFOUR-Cu-dpds, and 2162071 for $C_3H_6$-loaded BFFOUR-Cu-dpds. All other data needed to evaluate the conclusions are in the main text or the supplementary materials. Source data are provided with this paper.

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

## Acknowledgements

This study was supported by the National Natural Science Foundation of China (No. 22322807, 22168023, and 22268029) and the Natural Science Foundation of Jiangxi Province (No. 20224ACB204003).

## Author contributions

Y.P. designed the experiments and conducted sample synthesis, characterization, data processing, graph drawing, and article writing. H.X. carried out density functional theory (DFT) calculations, data collection

and processing. P.Z. carried out density functional theory (DFT) calculations. Z.W.Z. conducted sample synthesis and data collection. X.L. conducted sample synthesis, data collection, and article revision. S.T. conducted sample synthesis and data collection. Y.L. conducted sample synthesis and data collection. Z.L.Z. conducted sample synthesis and data collection. W.Z. conducted data collection and graph drawing. Z.D. conducted data collection. J.L. conducted sample synthesis and data collection. Y.Z. conducted draw graphs and collect data. Z.W. draw graphs and collected data. J.C. revised the article and draw charts. Z.Y. Z. revised the article and draw charts. S.C. revised the article and draw charts. S.D. revised the article. J.W. conceived the experiment, and conducted the article revision, writing and submission.

## Competing interests

The authors declare no competing interests.
