## [Peer Review File · Nature Communications]

Interaction-Selective Molecular Sieving Adsorbent for Direct Separation of Ethylene from Senary C2-C4 Olefin/Paraffin MixtureREVIEWER COMMENTS

Reviewer #1 (Remarks to the Author):

Authors synthesized a new MOF BFFOUR-Cu-dpds that exhibits an interesting interaction-selective molecular sieving mechanism for separating ethylene from olefin/paraffin mixtures. The study appears to be careful and complete, and mechanism/results are novel and intriguing. I suggest authors to address following comments and questions.

- 1) Authors mentioned the energy costs to produce polymer-grade C₂H₄ in introduction. Can authors estimate the cost of using the BFFOUR-Cu-dpds adsorbent for purifying C₂H₄? What is the cost for fabricating BFFOUR-Cu-dpds?
- 2) The flexible MOF that allows interaction-selective sieving (Fig. 1A) is interesting. In the molecular simulation studies, did authors observe the lattice-structure change after introducing C₂H₄ into the porous sorbent.
- 3) Related to #2, how did the lattice structure change when calculating the energy landscape? It is surprising that energy barriers for paraffins are so high (several eV). Did authors use an enlarged lattice structure for olefins and the original crystal structure for paraffins?
- 4) Recent studies also show the similar MOF with anionic pillars (e.g SIFSIX-Cu-TPBDA) are good for acetylene purification. Can BFFOUR-Cu-dpds be used for acetylene purification as well?
- 5) Water is also present in most operations and sometimes can damage MOFs. Will water affect the performance of BFFOUR-Cu-dpds when purifying C₂H₄?

Reviewer #2 (Remarks to the Author):

The authors developed a new 3D Cu-based MOF denoted to BAFFOUR-Cu-dpds, enabling to discriminating of the molecules between olefins and paraffin molecules of C₂–C₄ coupled with high separation performance. The characterization and assessment of separation performance results were well organized and explained logically. This adsorption characteristic is highly promising for separating the olefins from olefins/paraffins gas mixture with an adsorption-based separation process. In addition, this MS will be interested in the relevant MOF and separation research field. Thus, the reviewer recommends the publication of "Nat. Commun. Journal" after addressing minor points as follows:

1. Why was the stability test examined only in the pH ranges of 5–13? How about the stability at lower pH less than 5? In addition, the used acid and base solvent should be provided more detail in SI.
2. In the fitting results for C₂-C₃ paraffins with the DSLF model, the minus values of N₂MAX are not acceptable. The authors should use the appropriate fitting model or fitting results to ensure the validity of IAST selectivity values from the model-fitted parameters.
3. There is no experimental procedure for assessing the kinetic measurement for C₂-C₄ olefins and paraffins. The details of kinetic measurement should be provided.

4. In the SI, the authors mention the calculation of "static binding energy", like the case of ref. 36. Please explain the related results in the MS.
5. The authors mention that the CASTEP module was used to calculate the static binding energy (in the SI), and "energy barrier" was calculated using the CASTEP module in ref. 36. However, in the SI, the authors mention that the DMol3 module was used to calculate the energy barrier. Is there a particular reason why the authors used not CASTEP but DMol3 for energy barrier calculation? And, computational details related to energy barrier calculation in the SI seem to be not for DMol3 but for CASTEP, given the energy unit (eV). Please clarify the computational details.
6. The authors performed the isotherm and kinetic adsorption measurement with equimolar paraffin/olefin gas mixture to present evidence for the exclusive adsorption of paraffins (Fig. S26–27). However, these experiments are unsuitable evidence for rejecting paraffin adsorption. To provide direct evidence for adsorption only olefins, the analysis of bulk gas composition should be provided while measuring adsorption for both isotherm and kinetic adsorption. The reviewer thought that providing adsorption and desorption profiles in selected representative conditions was enough to prove the absence of co-adsorption between olefins/paraffins. For this consideration, desorption profiles in Fig S39a-c, C₂H₆, C₃H₈, n-C₄H₁₀ were checked, and the paraffins (C₂–C₄) were concurrently detected in the outlet stream during He purges at 333 K. Consequently, the partial co-adsorption of paraffins occurred while measuring the dynamic adsorption of C₂–C₄ gas mixtures.
7. The schematic of the breakthrough set-up in Fig S38 was inconsistent with explaining the transient breakthrough experiment in the SI part (MS or GC detector).
8. How were the dynamic selectivity values of olefins/paraffins compared with those of IAST prediction for C₂H₄/C₂H₆, C₃H₆/C₃H₈, and n-C₄H₈/n-C₄H₁₀?
9. In the multi-cyclic breakthrough and isotherm measurements (Fig. S45–48), each experiment's regeneration condition (e.g., temperature, flow rate of He, and regeneration time) was unclear.
10. Minor points need to be revised as follows:
 - a) The denotation for "Relative pressure (P/P₀)" of the x-axis in all isotherm curves for olefins and paraffins should be revised to Pressure (bar).
 - b) Miss a citation of Table S2 in the sentences "These adsorption enthalpy values were notably lower than many leading olefin-selective adsorbents (Table S2), such as PAF-1-SO₃Ag (106 kJ mol⁻¹ for C₂H₄), CuI@UiO-66-(COOH)₂ (48.5 kJ mol⁻¹ for C₂H₄), Co-gallate (41 kJ mol⁻¹ for C₃H₆), and KAUST-7 (57.4 kJ mol⁻¹ for C₃H₆), suggesting the easy regenerations for BFFOUR-Cu-dpds."
 - c) Miss a citation of Figure 5E–F in the sentences "As shown in Figure 5E, C₂H₄ molecules were rapidly adsorbed within 3 min, as evidenced by the appearance of stretching vibrations of $\nu_{as}(\text{CH}_2)$ at 2987~3124 cm⁻¹, $\nu(\text{C}=\text{C})$ at 1608 cm⁻¹, $\delta(\text{CH}_2)$ at 1444 cm⁻¹, $\sigma(\text{CH}_2)$ at 948 cm⁻¹, and $\omega(\text{CH}_2)$ at 825 cm⁻¹ of adsorbed C₂H₄.", and Similarly, the adsorption equilibrium of C₃H₆ was attained within 9 min in BFFOUR-Cu-dpds and could be easily regenerated under the same conditions within 12 min (Figure 5F).
 - d) In the legend of Figure S1, the unit of kinetic diameter difference between C₂ molecules "0.028 nm Å" needs to be corrected to "0.028 nm."
 - e) In the legend of Figure S43, the sample name should be changed from BFFOUR-cu-dpds to GBC-900.

Reviewer #3 (Remarks to the Author):

This paper shows a flexible 2D fluorinated MOF (BFFOUR-Cu-dpds) for olefin/paraffin separation via molecular sieving mechanism. The simultaneous separation of C2-C4 olefins from C2-C4 paraffins is an interesting result. However, this is a simple extension of previous works for flexible 2D fluorinated MOFs (SiFSIX-Cu-dps and ZUL-100), which have almost similar structures with BFFOUR-Cu-dpds and showed C2-C3 hydrocarbon separation via molecular sieving mechanism. Overall, I cannot find sufficient novelty and scientific insight deserved to be published in this high impact journal. For the above reasons, it is regrettable that I cannot recommend this paper to be published in Nature Communications.

Response to Reviewers' Comments

Title: Interaction-Selective Molecular Sieving Adsorbent for Direct Separation of Ethylene from Senary C2-C4 Olefin/Paraffin Mixture

Manuscript ID: NCOMMS-23-35385-T

Corresponding Author: Prof. Jun Wang

COMMENTS TO AUTHOR:

Reviewer #1 (*Remarks to the Author*): *Authors synthesized a new MOF BFFOUR-Cu-dpds that exhibits an interesting interaction-selective molecular sieving mechanism for separating ethylene from olefin/paraffin mixtures. The study appears to be careful and complete, and mechanism/results are novel and intriguing. I suggest authors to address following comments and questions.*

Author Response: We thank Reviewer #1 for the positive and valuable comments.

Comment 1. *Authors mentioned the energy costs to produce polymer-grade C₂H₄ in introduction. Can authors estimate the cost of using the BFFOUR-Cu-dpds adsorbent for purifying C₂H₄? What is the cost for fabricating BFFOUR-Cu-dpds?*

Author Response: Thank you for the valuable comment. The adsorptive separation can save ~60% energy consumption compared to the predominate distillation method (DOE report. Materials for separation technology energy and emission reduction opportunities, 2005). According to the current price of dpds ligand (\$15.08 per gram), the overall production price for BFFOUR-Cu-dpds is approximately \$12.52 per gram. Notably, the cost will be significantly reduced in large-scale production of dpds and BFFOUR-Cu-dpds. Moreover, MOF adsorbents are a better choice that are capable for polymer-grade C₂H₄ production compared to porous carbons and zeolites. Meanwhile, the MOF adsorbents in pressure swing adsorption (PSA) columns can be cycled thousands of times, undoubtedly reducing the application cost. We have also elaborated on the stability of the material in detail in the manuscript.

Comment 2. *The flexible MOF that allows interaction-selective sieving (Fig. 1A) is interesting. In the molecular simulation studies, did authors observe the lattice-structure charge after introducing C₂H₄ into the porous sorbent.*

Author Response: Thank you for the comment. We have calculated the charge density bias plots before and after introducing gas molecules (Figure S37). In Figure S37a, there are apparent negative charge transfers from F atom in BF₄⁻ anions to H atoms in C₂H₄ molecule.

Figure S37. Charge density bias plots showing the interactions between (a) C_2H_4 , (b) C_3H_6 , (c) $n-C_4H_8$ and the framework. Color code: F, green; Cu, dark green; C, gray; H (in framework), light gray; H (in hydrocarbons), dark purple; N, blue. S, yellow. Charge distribution: positive charge, yellow; negative charge, blue.

Comment 3. *Related to #2, how did the lattice structure change when calculating the energy landscape? It is surprising that energy barriers for paraffins are so high (several eV). Did authors use an enlarged lattice structure for olefins and the original crystal structure for paraffins?*

Author Response: Thank you for the valuable comment. For the calculation of transition states, the activated structure was applied for all calculations with $1 \times 1 \times 2$ superlattices. For gas-loaded models, we introduce a molecule into a single lattice by locating the task in the sorption module, then expand the lattice to $1 \times 1 \times 2$ superlattices and complete structural relaxation by the Dmol3 module. Such high energy barriers indicate that the paraffin molecules are difficult to diffuse into the frameworks, consistent with the experimental data.

Comment 4. *Recent studies also show the similar MOF with anionic pillars (e.g SIFSIX-Cu-TPBDA) are good for acetylene purification. Can BFFOUR-Cu-dpds be used for acetylene purification as well?*

Author Response: Thank you for the valuable suggestion. The C_2H_2 purification (C_2H_2/CO_2) is important in the chemical industry. As recommended, the adsorption isotherms for C_2H_2 and CO_2 were collected at 298 K (Figure R1). The BFFOUR-Cu-dpds adsorbent exhibited a higher C_2H_2 adsorption capacity ($35.6 \text{ cm}^3 \text{ g}^{-1}$) compared to that of CO_2 ($15.0 \text{ cm}^3 \text{ g}^{-1}$), indicating the application potential for C_2H_2 purification. The calculated IAST selectivity for C_2H_2/CO_2 (50/50) was 111.4, surpassing many top-ranking MOF adsorbents.

Figure R1. The adsorption isotherms for C₂H₂ and CO₂ at 298 K.

Comment 5. Water is also present in most operations and sometimes can damage MOFs. Will water affect the performance of BFFOUR-Cu-dpds when purifying C₂H₄?

Author Response: Thank you for the valuable comment. We totally agree that water vapor is inevitable in industrial gas streams. The separation performances under humid conditions (RH = 61.9%) have been evaluated. As shown in Figure S39, the breakthrough curve for C₂H₄/C₂H₆ separation demonstrated the efficient C₂H₄/C₂H₆ separation under humid conditions despite the slightly decreased retention time.

Figure S39. Breakthrough curves for C₂H₄/C₂H₆ (0.5/0.5, v/v, 1.0 ml min⁻¹ and 298 K) mixture in dry and humid conditions.

Modifications:

Manuscript: Page 13

For the gas-mixture of C₂H₄/C₂H₆ (0.5/0.5, v/v), C₂H₆ was rapidly eluted from the column at a flow rate of 1.0 ml min⁻¹, while C₂H₄ exhibited substantial retention in the column for 28 min until saturation (Figure 5A). Notably, the efficient C₂H₄/C₂H₆ separation was also obtained despite slightly decreased retention time under humid conditions (RH = 61.9%).

Manuscript: Page 19

For the breakthrough experiments with water vapor, the gas mixture passed through a water vapor saturator under ambient conditions. The humidity was measured by a relative humidity meter (UT333, UNI-T, China) at the inlet of the column. Following each breakthrough experiment, the sample was desorbed *in-situ* in the column with a He sweeping of 5 mL min⁻¹ at 333 K for 2 hours and attained high-purity olefin gas (99.5%).

Reviewer #2 (Remarks to the Author): *The authors developed a new 3D Cu-based MOF denoted to BFFOUR-Cu-dpds, enabling to discriminating of the molecules between olefins and paraffin molecules of C₂–C₄ coupled with high separation performance. The characterization and assessment of separation performance results were well organized and explained logically. This adsorption characteristic is highly promising for separating the olefins from olefins/paraffins gas mixture with an adsorption-based separation process. In addition, this MS will be interested in the relevant MOF and separation research field. Thus, the reviewer recommends the publication of “Nat. Commun. Journal” after addressing minor points as follows:*

Author Response: We thank Reviewer #2 for the positive and constructive comments.

Comment 1. *Why was the stability test examined only in the pH ranges of 5–13? How about the stability at lower pH less than 5? In addition, the used acid and base solvent should be provided more detail in SI.*

Author Response: Thank you for the valuable comment. The stability of adsorbents is a critical parameter in industry applications. As recommended, we conducted the stability tests in aqueous solutions with pH=1 and 3. As shown in Figure S13, the intensity of characteristic PXRD peaks significantly decreased, indicating its partial structure decomposition. Detailed information for used acid (hydrochloric acid) and base (sodium hydroxide) solvents with various concentrations has been added in the revised *Manuscript*.

Modifications:

Manuscript: Page 17

Structure stability Tests.

To evaluate the solvent stability, 100 mg BFFOUR-Cu-dpds were placed separately in 20 mL vials with 15 mL of different organic and pH solvents for 7 days. The resulting solid was filtered, activated at 80 °C for 12 hours, and characterized by PXRD and adsorption analysis. The used acid was hydrochloric acid solution and sodium hydroxide solution with various concentrations.

Comment 2. *In the fitting results for C2-C3 paraffins with the DSLF model, the minus values of N2MAX are not acceptable. The authors should use the appropriate fitting model or fitting results to ensure the validity of IAST selectivity values from the model-fitted parameters.*

Author Response: Thank you for pointing out the issue. We have re-calculated and confirmed the fitting parameters. The corresponding IAST selectivities have been updated, the C₂H₄/C₂H₆ selectivity increased from 28.4 to 68.8, and the C₃H₆/C₃H₈ selectivity increased from 103.6 to 108.4.

Modifications:

Supporting information: Page 24

Figure S23. DSLF fitting curves for (a) C_2H_4 and (b) C_2H_6 , (c) C_3H_6 and (d) C_3H_8 , and (e) $n-C_4H_8$ and (f) $n-C_4H_{10}$ adsorption isotherms at 298 K on BFFOUR-Cu-dpds.

Supporting information: Page 58

Supplementary Table S2. The fitting parameters of the dual-site Langmuir-Freundlich equation model for C_2H_4 , C_2H_6 , C_3H_6 , C_3H_8 , $n-C_4H_8$, and $n-C_4H_{10}$ adsorption on BFFOUR-Cu-dpds. Dual-site Langmuir-Freundlich (DSLFL) model is listed below:

$$N = N_1^{\max} \times \frac{b_1 P^{C_1}}{1 + b_1 P^{C_1}} + N_2^{\max} \times \frac{b_2 P^{C_2}}{1 + b_2 P^{C_2}}$$

Adsorbates	N_1^{\max} (mmol g ⁻¹)	b_1 (bar ⁻¹)	C_1	N_2^{\max} (mmol g ⁻¹)	b_2 (bar ⁻¹)	C_2
C_2H_4 (298 K)	0.7686466	0.000192303	4.451723	0.7575	0.041505	1.032607
C_2H_6 (298 K)	2.266979	0.000687412	0.8198868	2.266979	0.000687412	0.8198868
C_3H_6 (298 K)	0.8419861	9.5289E-6	4.193398	0.5079988	2.427431E-4	2.064262
C_3H_8 (298 K)	0.0067056	1.217249	0.4009188	0.3302159	0.003252615	1.227404

n-C ₄ H ₈ (298 K)	0.87337	3.95E-3	1.35399	0.38638	1.41627E-12	9.38565
n-C ₄ H ₁₀ (298 K)	1.55635	2.78E-3	0.66162	6.15E-3	1.46165	2.34353

Manuscript: Page 7

Specifically, BFFOUR-Cu-dpds showed a high IAST selectivity for C₂H₄/C₂H₆ (68.8), C₃H₆/C₃H₈ (108.4), and n-C₄H₈/n-C₄H₁₀ (22.9) at 298 K and 1.0 bar, surpassing many leading adsorbents such as Ni-gallate (16.8 for C₂H₄/C₂H₆), NOTT-300 (48.7 for C₂H₄/C₂H₆), ZnAtzPO₄ (12.4 for C₂H₄/C₂H₆), Fe₂(m-dobdc) (60 for C₃H₆/C₃H₈), and Fe₂(dobdc) (14.7 for C₃H₆/C₃H₈).

Comment 3. *There is no experimental procedure for assessing the kinetic measurement for C2-C4 olefins and paraffins. The details of kinetic measurement should be provided.*

Author Response: Thank you for the valuable comment. The experimental procedure for assessing the kinetic measurement was supplemented in the revised *Manuscript*.

Modifications:

Manuscript: Page 18

Kinetic adsorption measurements.

Intelligent Gravimetric Analyzer (IGA-100, HIDEN) was used to measure the time-dependent adsorption profiles of olefins and paraffins. About 80 mg of BFFOUR-Cu-dpds were subjected to evacuation under a high vacuum at 353 K for 24 hours before each adsorption measurement. After being cooled to a specific temperature, a single-component gas or olefin/paraffin gas-mixture was introduced into the chamber, and the mass of the sample loaded with gas molecules was continuously recorded for 80 min.

Comment 4. *In the SI, the authors mention the calculation of "static binding energy", like the case of ref. 36. Please explain the related results in the MS.*

Author Response: Thank you for the comment. As recommended, the static binding energy calculated by the DFT method have been supplemented in the revised *Manuscript* (Page 12). Static binding energy refers to the energy differences before and after adsorption at 0 K, representing the binding strength between adsorbents and adsorbates. The static binding energy was calculated to be 59.43 kJ mol⁻¹, 71.87 kJ mol⁻¹, and 82.52 kJ mol⁻¹ for C₂H₄, C₃H₆, and n-C₄H₈, respectively. The calculation method has also been modified in the revised *Manuscript*.

Modifications:

Manuscript: Page 12

The static binding energy was calculated to be 59.43 kJ mol⁻¹, 71.87 kJ mol⁻¹, and 82.52 kJ mol⁻¹ for C₂H₄, C₃H₆, and n-C₄H₈, respectively. The trend of binding strengths was consistent with the Q_{st} values.

According to the output files from the CASTEP modules, the “NB dispersion corrected est. 0 K energy* (Ecor-0.5TS)” was selected as the energy at 0 K.

Comment 5. *The authors mention that the CASTEP module was used to calculate the static binding energy (in the SI), and "energy barrier" was calculated using the CASTEP module in ref. 36. However, in the SI, the authors mention that the DMol3 module was used to calculate the energy barrier. Is there a particular reason why the authors used not CASTEP but DMol3 for energy barrier calculation? And, computational details related to energy barrier calculation in the SI seem to be not for DMol3 but for CASTEP, given the energy unit (eV). Please clarify the computational details.*

Author Response: Thank you for pointing out this mistake. We actually use the DMol3 module to calculate the diffusion energy barrier, because DMol3 has a faster calculation speed and relatively high calculation accuracy than CASTEP. The energy unit calculated by DMol3 is Ha (Hartree), and for a clearer representation, we converted it into eV unit according to 1 Ha=27.212 eV.

Modifications:

Manuscript: Page 21

The energy barrier calculations were carried out using the DMol3 module in Materials Studio. The periodic slab models with periodic boundary conditions were used to represent BFFOUR-Cu-dpds pore surface. The unit cells were optimized until the force acting between atoms was below 0.002 Ha/Å with SCF convergence of 10^{-6} . The Global orbital cutoff was 5.1 Å. The unit conversion followed the equation: 1 Ha=27.212 eV.

Comment 6. *The authors performed the isotherm and kinetic adsorption measurement with equimolar paraffin/olefin gas mixture to present evidence for the exclusive adsorption of paraffins (Fig. S26–27). However, these experiments are unsuitable evidence for rejecting paraffin adsorption. To provide direct evidence for adsorption only olefins, the analysis of bulk gas composition should be provided while measuring adsorption for both isotherm and kinetic adsorption. The reviewer thought that providing adsorption and desorption profiles in selected representative conditions was enough to prove the absence of co-adsorption between olefins/paraffins. For this consideration, desorption profiles in Fig S39a-c, C₂H₆, C₃H₈, n-C₄H₁₀ were checked, and the paraffins (C₂–C₄) were concurrently detected in the outlet stream during He purges at 333 K. Consequently, the partial co-adsorption of paraffins occurred while measuring the dynamic adsorption of C₂–C₄ gas mixtures.*

Author Response: Thank you for pointing out this issue. We agree that the gas-mixture isotherms cannot fully address the co-adsorption issue. While, some literature also used this as indirect evidence for

demonstrating this phenomenon (*Chem*, 2021, 7, 1006-1019; *Science*, 2016, 353,137-140; *J. Am. Chem. Soc.* 2020, 142, 41, 17795-17801). Furthermore, we also noticed that the immediate detection of C₂-C₄ paraffines in the desorption curve in Figures S40a-c, which could be ascribed to the surface adsorption on adsorbents or partial retention in quartz cotton at the inlet and outlet of adsorption column. To demonstrate this issue, we also carried out the breakthrough experiments with He/olefin/paraffin mixture (Figure S41), where He should not be adsorbed. The C₂-C₄ olefines concurrently eluted with He, suggesting the exclusion of C₂-C₄ paraffines.

Figure S41. Breakthrough curves for He/olefin/paraffin (0.2/0.4/0.4, v/v/v) mixture at 2.5 ml min⁻¹. Breakthrough curves for (a) He/C₂H₄/C₂H₆ (0.2/0.4/0.4, v/v/v) mixture, (b) He/C₃H₆/C₃H₈ (0.2/0.4/0.4, v/v/v) mixture, and (c) He/n-C₄H₈/n-C₄H₁₀ (0.2/0.4/0.4, v/v/v) mixture. The desorption conditions: 5 mL min⁻¹ He and 333 K.

Modifications:

Manuscript: Page 14

In addition, we introduced 20% inert He into olefin/paraffin gas-mixtures (He/olefin/paraffin, 0.2/0.4/0.4, v/v/v) to validate the absence of co-adsorption phenomenon during breakthrough experiments at a flow rate of 2.5 mL min⁻¹. As shown in Figure S41, the paraffins and He almost concurrently outflow from the column, indicating that the negligible paraffin adsorptions in gas-mixtures similar to He.

Comment 7. *The schematic of the breakthrough set-up in Fig S38 was inconsistent with explaining the transient breakthrough experiment in the SI part (MS or GC detector).*

Author Response: Thank you for pointing out this mistake. The detector is GC, which has been corrected in Figure S38.

Modifications:

Supporting information: Page 39

Figure S38. Schematic illustration of the setup for breakthrough experiments.

Comment 8. How were the dynamic selectivity values of olefins/paraffins compared with those of IAST prediction for C_2H_4/C_2H_6 , C_3H_6/C_3H_8 , and $n-C_4H_8/n-C_4H_{10}$?

Author Response: Thank you for the comment. The dynamic selectivity is also a critical parameter to evaluate the separation performances. The dynamic selectivity was calculated to be 9.16, 8.76, and 3.18 for equimolar C_2H_4/C_2H_6 , C_3H_6/C_3H_8 , and $n-C_4H_8/n-C_4H_{10}$ gas-mixtures, respectively. Due to the different calculation methods and indications, these values are not close to that of IAST predictions. The Ideal Adsorption Solution Theory (IAST) is a theory used to describe the adsorption isotherms of ideal mixed solutions/gases, which assumes that the adsorption of mixed solutions/gases is completely thermodynamic controlled (*AIChE J.*, 1965, 11: 121-127). While, the dynamic selectivity was determined by the adsorption capacity ratio of olefins and alkanes in the purge gas (*Angew. Chem. Int. Ed.* 2021, 60, 22865-22870). Moreover, in adsorption columns, the adsorption of mixed gases is still controlled by adsorption kinetics. The similar phenomenon occurred in many olefin/paraffin sieving adsorbents, such as Co-gallate (*J. Am. Chem. Soc.* 2020, 142, 41, 17795–17801) and UTSA-280 (*Nat. Mater.* 2018, 17, 1128-1133). The corresponding dynamic selectivity has been added in the revised *Manuscript*.

Modifications:

Manuscript: Page 13

The dynamic selectivity was calculated to be 9.16, 8.76, and 3.18 for equimolar C_2H_4/C_2H_6 , C_3H_6/C_3H_8 , and $n-C_4H_8/n-C_4H_{10}$ gas-mixtures, respectively.

Manuscript: Page 19-20

Dynamic separation selectivity (α) was obtained by calculating the integral area of the desorption curves of alkanes and olefins during the blowing process. The calculation formula is as follows:

$$S_i = \int_{t_1}^{t_2} \left(1 - \frac{F}{F_0}\right) dt$$

$$\alpha = \frac{S_1/S_2}{y_1/y_2}$$

Here S_i is the integral area of the desorption curve of gas i , t_1 and t_2 represent the start and end times of desorption (min), F_0 and F are the inlet and outlet gas molar flow rates, y_1 and y_2 represent the mole fractions of 1 and 2.

Comment 9. *In the multi-cyclic breakthrough and isotherm measurements (Fig. S45–48), each experiment's regeneration condition (e.g., temperature, flow rate of He, and regeneration time) was unclear.*

Author Response: Thank you for the valuable suggestion. The regeneration conditions have been added into the legend of each figure.

Modifications:

Supporting information: Page 41

Figure S40. The breakthrough curve of (a) C₂H₄/C₂H₆ (0.5/0.5, v/v), (b) C₃H₆/C₃H₈ (0.5/0.5, v/v), and (c) n-C₄H₈/n-C₄H₁₀ (0.5/0.5, v/v) at 298 K; and desorption curve of (d) C₂H₄/C₂H₆ (0.5/0.5, v/v), (e) C₃H₆/C₃H₈ (0.5/0.5, v/v), and (f) n-C₄H₈/n-C₄H₁₀ (0.5/0.5, v/v). The signals of desorption conditions: 5 ml min⁻¹ He and 333 K for 12 h.

Supporting information: Page 47

Figure S46. Three breakthrough cycles curves of (a) C₂H₄/C₂H₆ (0.5/0.5, v/v), (b) C₃H₆/C₃H₈ (0.5/0.5, v/v), and (c) n-C₄H₈/n-C₄H₁₀ (0.5/0.5, v/v) on BFFOUR-Cu-dpds at 1 mL min⁻¹ and 298 K. The signals of desorption conditions: 5 ml min⁻¹ He and 333 K for 90 min.

Supporting information: Page 48

Figure S47. Three breakthrough cycles for C₂H₄/C₂H₆/C₃H₆/C₃H₈/n-C₄H₈/n-C₄H₁₀ (0.167/0.167/0.167/0.167/0.167/0.167, v/v/v/v/v/v) on BFFOUR-Cu-dpds at 1.0 ml min⁻¹ and 298 K. The signals of desorption conditions: 5 ml min⁻¹ He and 333 K for 90 min.

Supporting information: Page 49

Figure S48. Three breakthrough cycles for C₂H₄/C₂H₆/C₃H₆/C₃H₈/n-C₄H₈/n-C₄H₁₀ (0.167/0.167/0.167/0.167/0.167/0.167, v/v/v/v/v/v) on BFFOUR-Cu-dpds at 2.0 mL min⁻¹ and 298 K. The signals of desorption conditions: 5 ml min⁻¹ He and 333 K for 90 min.

Supporting information: Page 50

Figure S49. Five repeated adsorption isotherms for (a) C₃H₆ and (b) n-C₄H₈ on BFFOUR-Cu-dpds at 298 K and 1 bar. The signals of desorption conditions: vacuum 333 K for 12 h.

Comment 10. *Minor points need to be revised as follows:*

a) *The denotation for “Relative pressure (P/P_0)” of the x-axis in all isotherm curves for olefins and paraffins should be revised to Pressure (bar).*

Author Response: Thank you for the valuable suggestion. As recommended, we have changed “ P/P_0 ” to “bar” in related figures.

b) *Miss a citation of Table S2 in the sentences “These adsorption enthalpy values were notably lower than many leading olefin-selective adsorbents (Table S2), such as PAF-1-SO₃Ag (106 kJ mol⁻¹ for C₂H₄), Cu^I@UiO-66-(COOH)₂ (48.5 kJ mol⁻¹ for C₂H₄), Co-gallate (41 kJ mol⁻¹ for C₃H₆), and KAUST-7 (57.4 kJ mol⁻¹ for C₃H₆), suggesting the easy regenerations for BFFOUR-Cu-dpds.”*

Author Response: Thank you for the valuable suggestion. As recommended, we have cited the comparison table in this sentence (Tables S4 and S5), if they are not correct. The Table S2 is the table for fitting parameters.

Modifications:

Manuscript: Page 8

These adsorption enthalpy values were notably lower than many leading olefin-selective adsorbents (Tables S4 and S5), such as PAF-1-SO₃Ag (106 kJ mol⁻¹ for C₂H₄), Cu^I@UiO-66-(COOH)₂ (48.5 kJ mol⁻¹ for C₂H₄), Co-gallate (41 kJ mol⁻¹ for C₃H₆), and KAUST-7 (57.4 kJ mol⁻¹ for C₃H₆), suggesting the easy regenerations for BFFOUR-Cu-dpds.

c) *Miss a citation of Figure 2E–F in the sentences “As shown in Figure 2E, C₂H₄ molecules were rapidly adsorbed within 3 min, as evidenced by the appearance of stretching vibrations of $\nu_{a,as}(CH_2)$ at 2987~3124 cm⁻¹, $\nu(C=C)$ at 1608 cm⁻¹, $\delta(CH_2)$ at 1444 cm⁻¹, $\sigma(CH_2)$ at 948 cm⁻¹, and $w(CH_2)$ at 825 cm⁻¹ of adsorbed C₂H₄.”, and Similarly, the adsorption equilibrium of C₃H₆ was attained within 9 min in BFFOUR-Cu-dpds and could be easily regenerated under the same conditions within 12 min (Figure 5F).*

Author Response: Thank you for the valuable suggestion. As recommended, we have modified the sentence. Similarly, the adsorption equilibrium of C₃H₆ was attained within 9 min in BFFOUR-Cu-dpds and could be easily regenerated under the same conditions within 12 min (Figure 2F).

Modifications:

Manuscript: Page 8

As shown in Figure 2e, C₂H₄ molecules were rapidly adsorbed within 3 min, as evidenced by the appearance of stretching vibrations of $\nu_{a,as}(CH_2)$ at 2987~3124 cm⁻¹, $\nu(C=C)$ at 1608 cm⁻¹, $\delta(CH_2)$ at 1444 cm⁻¹, $\sigma(CH_2)$ at 948 cm⁻¹, and $w(CH_2)$ at 825 cm⁻¹ of adsorbed C₂H₄.

d) *In the legend of Figure S1, the unit of kinetic diameter difference between C2 molecules “0.028 nm Å” needs to be corrected to “0.028 nm.”*

Author Response: Thank you for pointing out this mistake. We have corrected this mistake.

Modifications:

Supporting information: Page 2

Figure S1. Molecular dimensions of (a) C₂H₄ and (b) C₂H₆ (kinetic diameter difference: 0.028 nm, boiling point (b.p) difference: 15 K);

e) In the legend of Figure S43, the sample name should be changed from BFFOUR-cu-dpds to GBC-900.

Author Response: Thank you for pointing out this mistake. We have corrected the error.

Modifications:

Supporting information: Page 45

Figure S44. Adsorption isotherms of C₂H₄, C₃H₆, and n-C₄H₈ on GBC-900 at 298 K: (a) logarithmic model and (b) linear model.

Reviewer #3 (Remarks to the Author):

Comment: This paper shows a flexible 2D fluorinated MOF (BFFOUR-Cu-dpds) for olefin/paraffin separation via molecular sieving mechanism. The simultaneous separation of C₂-C₄ olefins from C₂-C₄ paraffins is an interesting result. However, this is a simple extension of previous works for flexible 2D fluorinated MOFs (SiFSIX-Cu-dpds and ZUL-100), which have almost similar structures with BFFOUR-Cu-dpds and showed C₂-C₃ hydrocarbon separation via molecular sieving mechanism. Overall, I cannot find sufficient novelty and scientific insight deserved to be published in this high impact journal. For the above reasons, it is regrettable that I cannot recommend this paper to be published in Nature Communications.

Author Response: Thanks for Reviewer 3's evaluation and comment. We totally agree that the anion-pillared MOFs are under intensive investigations, and many research groups have reported excellent results based on this type of material. However, we do report some advances in this work, not only a simple extension of previous works. First, unlike reported anion-pillared MOFs, we have decoupled the assembled component to

one-dimensional $\text{Cu}(\text{dpds})_2(\text{BF}_4)_2$ chains, which are further interwoven by angular BF_4^- anions. This unprecedented topology is never reported (**2CI** by Topos program). Second, the molecule-sieving mechanism is based on different interactions rather than molecule sizes. Therefore, the as-synthesized BFFOUR-Cu-dpds can simultaneously sieve $\text{C}_2\text{-C}_4$ olefins from corresponding paraffins, which is also never reported in anion-pillared MOFs. Third, based on the excellent performances, a single adsorption column filled with BFFOUR-Cu-dpds can separate olefins and paraffins from six-component $\text{C}_2\text{-C}_4$ gas-mixtures. Then, the second adsorption column filled with porous carbons can facily isolate C_2H_4 , C_3H_6 , and $\text{n-C}_4\text{H}_8$ according to different carbon atoms. Therefore, high-purity C_2H_4 (>99.99%) can be directly produced *via* a two-column configuration from six-component $\text{C}_2\text{-C}_4$ olefins and paraffins mixtures. The novel operation flow can significantly streamline the traditional flow with multiple distillation towers. I hope the response has clearly demonstrated the advances and advantages of BFFOUR-Cu-dpds.

REVIEWERS' COMMENTS

Reviewer #1 (Remarks to the Author):

Authors have addressed my questions adequately. I recommend its publication.

Reviewer #2 (Remarks to the Author):

The contribution by Yong Peng and co-workers entitled "Interaction-Selective Molecular Sieving Adsorbent for Direct Separation of Ethylene from Senary C2-C4 Olefin/Paraffin Mixture" demonstrates the remarkable performance for separating olefins with BFFOUR-Cu-dpds adsorbent. The revised manuscript is well constructed, and all issues raised by the reviewer are addressed. I am delighted to recommend it for publication in Nat. Commun. Journal.

Reviewer #3 (Remarks to the Author):

I rejected the publication of this paper in the first review due to the novelty issue. But, I changed my mind considering the reviews of other reviewers and the responses from the authors. However, there are still unclear and questionable issues. I recommend the publication of this paper if the authors clearly address the following issues:

In this study, the IAST was applied to calculate the adsorption selectivity. But, as the authors answered to the Reviewer #'s comment, the IAST assumes that the adsorption of mixed solutions/gases is completely thermodynamic controlled. Since the adsorbent in this study shows slow adsorption kinetics, especially for C₃H₆, the application of the IAST is not proper. Actually, due to this reason, the IAST selectivity values of BFFOUR-Cu-dpds were much lower than the breakthrough selectivity. Since the calculated IAST selectivity may have been largely overestimated, it should not be compared with the benchmark adsorbents. I think the comparisons of the uptake ratios are also not proper since it does not consider the kinetic effect.

Rather than that, the breakthrough selectivity, which is more meaningful, should be compared with those of the benchmark adsorbents.

The stability tests under pH values were just done by PXRD. But, there were clear decreases in the intensities of the PXRD peaks, especially at the low angle peaks. Therefore, the BET surface areas of the samples before and after the pH treatment should be also provided.

Fig. S17b: A hysteresis is observed in C₃H₆ isotherm. Some explanations should be provided.

Fig. S18: While C2 and C3 adsorption isotherms were measured at 273 K and 323 K (Figs. S16-S17), C4 adsorption isotherms were measured at 283 K and 313 K. Why?

Fig. S19: It is hard to see the differences among the isotherms. To see the differences, the scale of y axis should be adjusted.

Fig. S20: Since the transient adsorption curves were provided, the diffusional time constants (D/r^2) should be calculated using a proper model to analyze the kinetic data more clearly.

Fig. S29: The trends of the Q_{st} values with gas loadings are different among the three gases. Some explanations should be provided.

Reviewer #1 (Remarks to the Author):

Authors have addressed my questions adequately. I recommend its publication.

Author Response: We thank reviewer #1 for the valuable comments and positive recommendation.

Reviewer #3 (Remarks to the Author):

I rejected the publication of this paper in the first review due to the novelty issue. But, I changed my mind considering the reviews of other reviewers and the responses from the authors. However, there are still unclear and questionable issues. I recommend the publication of this paper if the authors clearly address the following issues:

In this study, the IAST was applied to calculate the adsorption selectivity. But, as the authors answered to the Reviewer #'s comment, the IAST assumes that the adsorption of mixed solutions/gases is completely thermodynamic controlled. Since the adsorbent in this study shows slow adsorption kinetics, especially for C₃H₆, the application of the IAST is not proper. Actually, due to this reason, the IAST selectivity values of BFFOUR-Cu-dpds were much lower than the breakthrough selectivity. Since the calculated IAST selectivity may have been largely overestimated, it should not be compared with the benchmark adsorbents. I think the comparisons of the uptake ratios are also not proper since it does not consider the kinetic effect.

Rather than that, the breakthrough selectivity, which is more meaningful, should be compared with those of the benchmark adsorbents.

Author Response: We thank reviewer #3 for acknowledging our efforts for revising our manuscript and providing valuable suggestions. We agree that the IAST selectivity and uptake ratio have deficiency in determining practical separation performances. However, considering that these two parameters are commonly used in most published papers for comparison purposes, we are inclined to retain them while adding additional explanations in the revised manuscript. Furthermore, as recommended, we have added the comparison of breakthrough selectivity.

Modifications:

Manuscript: Page 13

Considering that the IAST selectivity and uptake ratio are determined by equilibrium effect, the dynamic selectivity based on breakthrough curves was calculated to be 9.16, 8.76, and 3.18 for equimolar C₂H₄/C₂H₆, C₃H₆/C₃H₈, and n-C₄H₈/n-C₄H₁₀, respectively. These values demonstrate comparable performance to top-ranking adsorbents, such as NUS-6(Hf)-Ag (4.4 for C₂H₄/C₂H₆),⁷¹ ZJU-75a (14.7 for C₃H₆/C₃H₈),⁷² Y-abtc (8.3 for C₃H₆/C₃H₈),⁷³ and KAUST-7 (12.0 for C₃H₆/C₃H₈).⁸

Comment 1: The stability tests under pH values were just done by PXRD. But, there were clear decreases in the intensities of the PXRD peaks, especially at the low angle peaks. Therefore, the BET surface areas of the samples before and after the pH treatment should be also provided.

Author Response: Thank you for the valuable comment. As recommended, the BET specific surface of BFFOUR-Cu-dpds were measured after immersing in various pH solutions for 7 days (Supplementary Figure 13c). Notably, the framework exhibited complete decomposition at pH values ≥ 13 and pH ≤ 3 , which corroborated the XRD results.

Modifications:

Supporting information: Page S20

Supplementary Figure 13. PXRD patterns and CO₂ adsorption isotherms at 195 K after various treatments. PXRD patterns after various treatments in (a) water and organic solvents, (b) aqueous solutions at various pH values for 7 days, and (c) CO₂ adsorption isotherms at 195 K after immersing in various pH aqueous solutions for 7 days.

Modifications:

Manuscript: Page 6

Furthermore, the PXRD characteristic peaks and BET specific surface areas measured at 195 K of BFFOUR-Cu-dpds almost remained intact after soaking in various organic solvents and acidic/basic aqueous solutions (pH=5-11) for 7 days, indicating its excellent structure and chemical stability (Supplementary Figure 13).

Comment 2: Fig. S17b: A hysteresis is observed in C₃H₆ isotherm. Some explanations should be provided.

Author Response: Thank you for the comment. BFFOUR-Cu-dpds exhibits remarkable structural flexibility. The activated BFFOUR-Cu-dpds has a pore diameter of 3.8 Å (Supplementary Figure 15), which is smaller than the kinetic diameters of these three molecules. Consequently, the adsorption process inevitably induces changes in the framework structure of BFFOUR-Cu-dpds, which serves as the primary reason behind the observed adsorption hysteresis effect. This phenomenon is also observed in other flexible materials such as UTSA-300 (J. Am. Chem. Soc. **2017**, 139, 8022-8028), NCU-100 (J. Am. Chem. Soc. **2020**, 142, 9744-9751), and Co(4-DPDS)₂CrO₄ (Adv. Sci. **2023**, 10, 2207127).

Comment 4: Fig. S18: While C2 and C3 adsorption isotherms were measured at 273 K and 323 K (Figs. S16-S17), C4 adsorption isotherms were measured at 283 K and 313 K. Why?

Author Response: Thank you for the comment. As shown in Supplementary Figure 18, the channel opening pressure of n-butene is approximately 0.05 bar at 283 K, while it increases to above 0.4 bar at 313 K. This substantial disparity in opening pressure leads to distinct adsorption isotherm shapes. If the testing temperature was too low, the gate-opening phenomenon may be disappeared and causing significant calculation errors. Therefore, we chose to measure the adsorption isotherms at 283 K, 298 K, and 313 K.

Comment 5: Fig. S19: It is hard to see the differences among the isotherms. To see the differences, the scale of y axis should be adjusted.

Author Response: Thank you for the comment, we have made modifications to Figure S19.

Modifications:

Supporting information: Page 26

Supplementary Figure 19. Adsorption isotherms of C₂H₆, C₃H₈, and n-C₄H₁₀ are at different temperatures. (a) Adsorption isotherms of C₂H₆, C₃H₈, and n-C₄H₁₀ at 298 K; (b) adsorption isotherms of C₂H₆ and C₃H₈ at 323 K and n-C₄H₁₀ at 313 K on BFFOUR-Cu-dpds.

Comment 6: Fig. S20: Since the transient adsorption curves were provided, the diffusional time constants (D/r^2) should be calculated using a proper model to analyze the kinetic data more clearly.

Author Response: Thanks for the valuable comment. As recommended, we calculated the diffusion time constants based on the kinetic adsorption curves.

Modifications:

Supporting information: Page S27

Supplementary Figure 20. The kinetic adsorption and fitting curves for C₂-C₄ at 298 K and 1 bar. (a) The kinetic adsorption curves and (b) fitting curves for C₂-C₄ olefins and paraffins at 1.0 bar.

Comment 7: Fig. S29: The trends of the Q_{st} values with gas loadings are different among the three gases. Some explanations should be provided.

Author Response: Thank you for the comment. The adsorption heat is determined by the binding energy between the adsorbate and the adsorbent (Science356, 2017, 1193-1196). Since BFFOUR-Cu-dpds is a flexible adsorbent, the adsorption processes involve obvious gate-opening phenomenon. The framework transformation is non-spontaneous and requires energy input for expansion. Consequently, at low adsorption amounts, the adsorption heat is relatively low. As more guest molecules are adsorbed, the framework opens up and provides accessible sites for guest molecules, leading to an increase in guest-host interactions. As a majority of the available sites become occupied, there is a corresponding decrease in adsorption heats.

REVIEWERS' COMMENTS

Reviewer #3 (Remarks to the Author):

The authors addressed all the comments from the reviewer. I think it is now acceptable for publication.